# PPTC7 antagonizes mitophagy by promoting BNIP3 and NIX degradation via SCF^FBXL4

Giang Thanh Nguyen-Dien [1,2,15], Brendan Townsend [1,15], Prajakta Gosavi Kulkarni [1,15], Keri-Lyn Kozul [1,12,15], Soo Siang Ooi[1], Denaye N Eldershaw [3], Saroja Weeratunga [3], Meihan Liu [3], Mathew JK Jones [4,5], S Sean Millard [1], Dominic CH Ng[1], Michele Pagano [6,7,8], Alexis Bonfim-Melo[4], Tobias Schneider [9,13], David Komander [9,13], Michael Lazarou [10,11,14], Brett M Collins [3✉] & Julia K Pagan [1✉]

## Abstract

**Mitophagy must be carefully regulated to ensure that cells maintain appropriate numbers of functional mitochondria. The SCF^FBXL4 ubiquitin ligase complex suppresses mitophagy by controlling the degradation of BNIP3 and NIX mitophagy receptors, and *FBXL4* mutations result in mitochondrial disease as a consequence of elevated mitophagy. Here, we reveal that the mitochondrial phosphatase PPTC7 is an essential cofactor for SCF^FBXL4-mediated destruction of BNIP3 and NIX, suppressing both steady-state and induced mitophagy. Disruption of the phosphatase activity of PPTC7 does not influence BNIP3 and NIX turnover. Rather, a pool of PPTC7 on the mitochondrial outer membrane acts as an adaptor linking BNIP3 and NIX to FBXL4, facilitating the turnover of these mitophagy receptors. PPTC7 accumulates on the outer mitochondrial membrane in response to mitophagy induction or the absence of FBXL4, suggesting a homoeostatic feedback mechanism that attenuates high levels of mitophagy. We mapped critical residues required for PPTC7–BNIP3/NIX and PPTC7-FBXL4 interactions and their disruption interferes with both BNIP3/NIX degradation and mitophagy suppression. Collectively, these findings delineate a complex regulatory mechanism that restricts BNIP3/NIX-induced mitophagy.**

**Keywords** Mitophagy; FBXL4; BNIP3; NIX; PPTC7
**Subject Categories** Autophagy & Cell Death; Post-translational Modifications & Proteolysis

## Introduction

Cells eliminate excessive or damaged mitochondria through mitophagy, a selective form of autophagy (Onishi et al, 2021; Uoselis et al, 2023). The upregulation of mitophagy receptors BNIP3 and NIX on the mitochondrial outer membrane acts as a signal to recruit the autophagosome (Hanna et al, 2012; Marinkovic and Novak, 2021; Rogov et al, 2017), in conditions such as hypoxia (Allen et al, 2013; Bellot et al, 2009; Zhao et al, 2020), and during the differentiation of specialised cell types like erythrocytes (Novak et al, 2010; Sandoval et al, 2008; Schweers et al, 2007), neurons (Esteban-Martinez et al, 2017; Ordureau et al, 2021), cardiomyocytes (Lampert et al, 2019; Zhao et al, 2020), keratinocytes (Simpson et al, 2021), and proinflammatory macrophages (Esteban-Martinez et al, 2017). It has long been known that BNIP3 and NIX expression is acutely upregulated by transcription (Sowter et al, 2001; Tracy et al, 2007), however, the molecular understanding of the mechanisms that restrict BNIP3 and NIX expression to prevent excessive mitophagy remains limited.

We and others have recently demonstrated that the abundance of BNIP3 and NIX receptors is regulated by the mitochondrially localised SCF^FBXL4 E3 ubiquitin ligase complex (Cao et al, 2023; Chen et al, 2023; Elcocks et al, 2023; Nguyen-Dien et al, 2023). FBXL4 is one of 69 F-box proteins that act as interchangeable substrate adaptors for SCF E3 ubiquitin ligase protein complexes. SCF^FBXL4 localises to the mitochondrial outer membrane and mediates the constitutive ubiquitylation and degradation of BNIP3 and NIX to limit their abundance and thereby suppress mitophagy. The FBXL4 gene is mutated in mtDNA Depletion Syndrome 13 (MTDPS13), a disease characterised by mitochondrial depletion caused by excessive mitophagy (Alsina et al, 2020; Bonnen et al, 2013; Gai et al, 2013).

[1]Faculty of Medicine, School of Biomedical Sciences, University of Queensland, Brisbane, QLD, Australia. [2]Department of Biotechnology, School of Biotechnology, Viet Nam National University-International University, Ho Chi Minh City, Vietnam. [3]The University of Queensland, Institute for Molecular Bioscience, Brisbane, QLD 4072, Australia. [4]The University of Queensland Frazer Institute, Faculty of Medicine, The University of Queensland, Brisbane, QLD 4102, Australia. [5]School of Chemistry & Molecular Biosciences, University of Queensland, Brisbane, QLD 4072, Australia. [6]Department of Biochemistry and Molecular Pharmacology, New York University Grossman School of Medicine, New York, NY 10016, USA. [7]Perlmutter Cancer Center, New York University Grossman School of Medicine, New York, NY 10016, USA. [8]Howard Hughes Medical Institute, New York University Grossman School of Medicine, New York, NY 10065, USA. [9]Walter and Eliza Hall Institute of Medical Research, Parkville, VIC, Australia. [10]Department of Biochemistry and Molecular Biology, Biomedicine Discovery Institute, Monash University, Melbourne, VIC 3068, Australia. [11]Department of Medical Biology, University of Melbourne, Melbourne, VIC 3068, Australia. [12]Present address: Department of Biochemistry and Molecular Biophysics, Washington University School of Medicine, MO 63110 St Louis, USA. [13]Present address: Department of Medical Biology, University of Melbourne, Melbourne, VIC 3068, Australia. [14]Present address: Walter and Eliza Hall Institute of Medical Research, Parkville, VIC, Australia. [15]These authors contributed equally: Giang Thanh Nguyen-Dien, Brendan Townsend, Prajakta Gosavi Kulkarni, Keri-Lyn Kozul. ✉E-mail: b.collins@imb.uq.edu.au; j.pagan@uq.edu.au

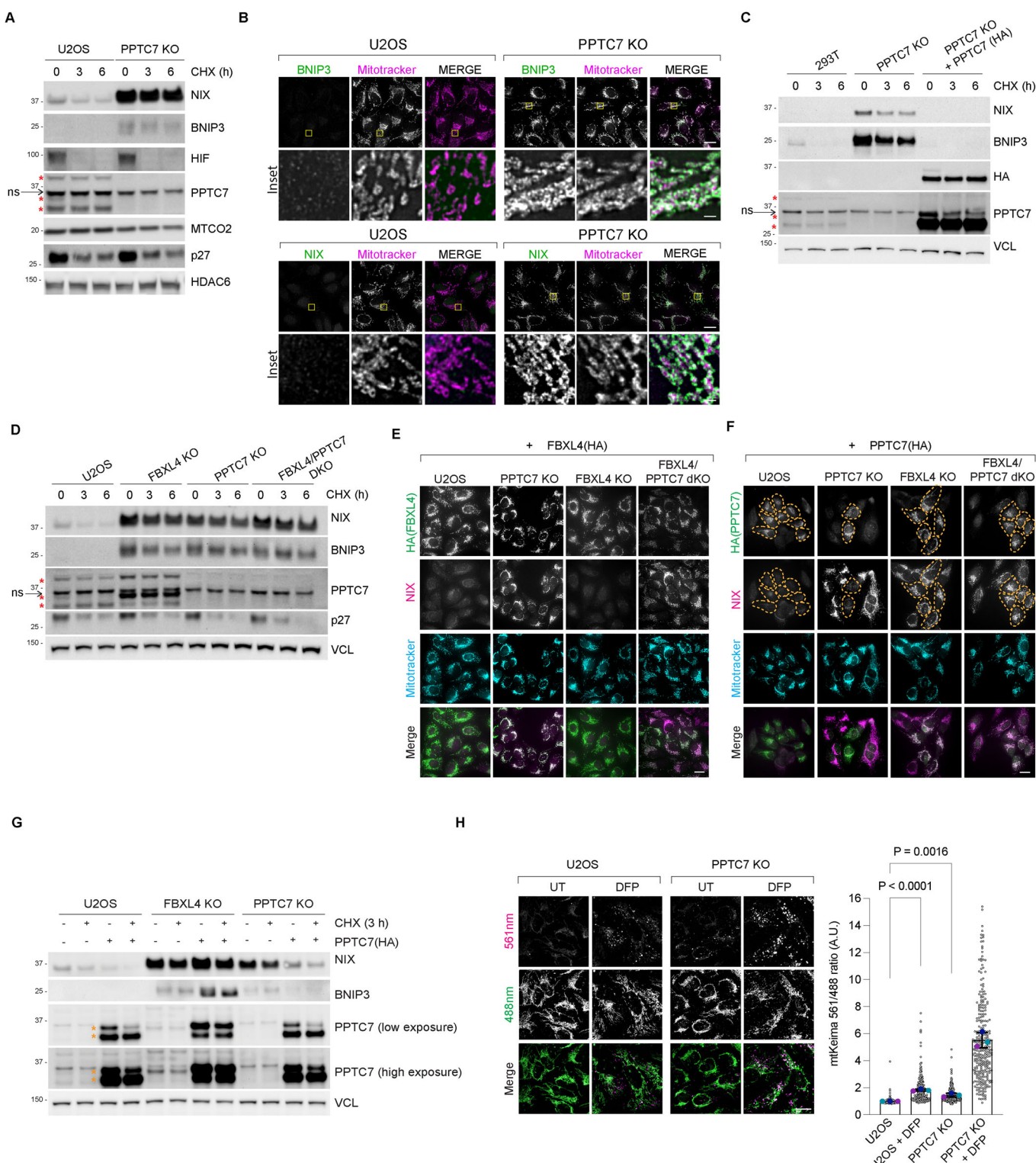

Little is known regarding the upstream mechanisms that regulate BNIP3 and NIX mitophagy receptor recognition via FBXL4. Phosphorylation or other types of modifications could potentially either enhance or sterically preclude the recognition of BNIP3 and NIX by FBXL4. Alternatively, a cofactor might be required to regulate the assembly of the SCF complex or to bridge the interaction between FBXL4 with BNIP3 and NIX.

The PP2C phosphatase PPTC7 (Protein Phosphatase Targeting COQ7) localises predominantly to the mitochondrial matrix (Guo et al, 2017; Niemi et al, 2019). Despite this localisation, it has been

**Figure 1.  PPTC7 and FBXL4 coordinate the turnover of BNIP3 and NIX mitophagy receptors to suppress mitophagy.**

(A) BNIP3 and NIX are stabilised in PPTC7-deficient U2OS cells. CRISPR-CAS9 was used to knockout PPTC7 in U2OS cells. Cells were treated with cycloheximide for the indicated times before immunoblotting as indicated. The red asterisks indicate PPTC7-specific bands at 28 kDa, 32 kDa and 40 kDa. ns = non-specific band at ~36 kDa. (B) Immunofluorescence staining of BNIP3 and NIX demonstrating increased levels of both proteins at mitochondria in PPTC7-deficient cells U2OS cells. Cells were stained with BNIP3 or NIX (in green) and counterstained with MitoTracker (in magenta). (C) Re-expression of PPTC7 into PPTC7-deficient cells reduces the stability of BNIP3 and NIX. PPTC7-HA was transduced into PPTC7-deficient 293T cells. The half-lives of BNIP3 and NIX were analysed by immunoblotting after a cycloheximide chase. (D) Targeting PPTC7 and FBXL4 simultaneously does not further increase BNIP3 and NIX levels or stability. PPTC7 was sequentially knocked out of previously generated FBXL4-deficient cells to make the double knockout of PPTC7 and FBXL4 (dKO FBXL4/PPTC7). The 32 kDa band of PPTC7 was upregulated in the FBXL4 KO cells. (E) FBXL4 requires PPTC7 to mediate the downregulation of NIX. FBXL4-HA was transduced in parental, FBXL4 KO, PPTC7 KO, and FBXL4/PPTC7 dKO cells. NIX protein levels (magenta) were monitored in FBXL4(HA)-expressing cells (green). Full figure is provided in Appendix Fig. S1. Scale bar = 20 microns. (F) PPTC7 requires FBXL4 to mediate the downregulation of NIX. PPTC7-HA was expressed in parental, FBXL4 KO, PPTC7 KO, and FBXL4/PPTC7 dKO cells. NIX protein levels (magenta) were monitored in PPTC7(HA) expressing cells (green). To correlate NIX levels with PPTC7 expression, PPTC7 expressing cells (green) are outlined in orange dotted lines. Full figure is provided in Appendix Fig. S1. Scale bar = 20 microns. (G) FBXL4 is required for the PPTC7-mediated downregulation of BNIP3 and NIX. PPTC7(HA) was transduced into U2OS, FBXL4 KO or PPTC7 KO cells and the half-lives of BNIP3 and NIX were monitored by immunoblotting. PPTC7(HA) rescue into PPTC7 KO cells causes a reduction of BNIP3 and NIX. In contrast, the expression of PPTC7(HA) into FBXL4 KO cells does not cause a reduction in either NIX or BNIP3. (H) PPTC7 deficiency leads to increased mitophagy in basal conditions and after DFP treatment. U2OS mt-Keima-PPTC7 KO cells were treated with DFP for 24 h. Emission signals at neutral pH were obtained after excitation with the 458 nm laser (green), and emission signals at acidic pH were obtained after excitation with the 458 nm laser 561 nm laser (magenta). Mitophagy is represented as the ratio of mt-Keima 561 nm fluorescence intensity to mt-Keima 458 nm fluorescence intensity for individual cells normalised to the mean of the untreated condition. Translucent grey dots represent measurements from individual cells. Coloured circles represent the mean ratio from independent experiments. The centre lines and bars represent the mean of the independent replicates +/− standard deviation. P values were calculated based on the mean values using a one-way ANOVA. n = 3. Data Information: (A, C, D) The red asterisks indicate PPTC7-specific bands at 28 kDa, 32 kDa and 40 kDa. ns = non-specific band at ~36 kDa. (G) The orange asterisks denote PPTC7(HA) expression. (B, E–H) Scale bar of main = 20 microns. (B) Scale bar of inset (yellow boxes) = 1 micron. Source data are available online for this figure.

identified as an interactor of BNIP3 and NIX (Huttlin et al, 2021; Huttlin et al, 2017; Luck et al, 2020; Pagliarini et al, 2008) and a suppressor of BNIP3 and NIX-dependent mitophagy (Niemi et al, 2023). Intriguingly, Pptc7 knockout mice exhibit phenotypes reminiscent of Fbxl4 knockout mice with decreased mitochondria content, increased mitophagy and severe metabolic defects which are associated with perinatal lethality (Alsina et al, 2020; Cao et al, 2023; Niemi et al, 2023; Niemi et al, 2019).

Here, we show that PPTC7 is a critical rate-limiting activator of FBXL4-mediated destruction of BNIP3 and NIX and is required to suppress excessive steady-state and mitophagy induced by pseudohypoxia. We propose that PPTC7 acts as an adaptor to enable the turnover of BNIP3 and NIX via the SCF$^{FBXL4}$. An outer membrane form of PPTC7 interacts with BNIP3 and NIX as well as with FBXL4 and these interactions are required for BNIP3 and NIX turnover and mitophagy suppression. We functionally validate in silico predictions of the PPTC7–NIX and PPTC7–FBXL4 interactions to reveal critical residues required for the assembly of the PPTC7–NIX/BNIP3 and PPTC7–FBXL4, and consequently for the turnover of BNIP3 and NIX as well as mitophagy suppression. Together, these findings provide a molecular understanding of the mechanisms that restrict NIX/BNIP3-stimulated mitophagy to prevent excessive mitophagy.

## Results

### PPTC7 and FBXL4 cooperate to mediate the turnover of BNIP3 and NIX

Corresponding with increased mitophagy, PPTC7-deficient cells and tissues have diminished steady-state levels of most mitochondrial proteins, except for BNIP3 and NIX, which instead exhibit significantly increased protein levels (Niemi et al, 2023; Niemi et al, 2019). To determine whether PPTC7 regulates BNIP3 and NIX expression at the level of protein stability, we generated PPTC7-

deficient cell lines using CRISPR/Cas9-mediated gene disruption. Successful knockout was confirmed by immunoblotting using an anti-PPTC7 antibody, which detected bands migrating at 28 kDa, 32 kDa and 40 kDa, specifically in parental cell lines but not in PPTC7 KO lines (Figs. 1A,D and EV1A–C). Cycloheximide chase assays demonstrated that PPTC7 deficiency significantly increased the levels of BNIP3 and NIX and resulted in the upregulation of BNIP3 and NIX at mitochondria, indicating that PPTC7 is required for BNIP3 and NIX turnover (Figs. 1A–C and EV1A,B). To assess whether PPTC7 impacts the transcriptional regulation of BNIP3 and NIX via HIF1α, we used the HIF1α inhibitor, echinomycin, to demonstrate that HIF1α inhibition did not abolish the accumulation of BNIP3 and NIX in PPTC7 KO cells (Fig. EV1C). The abnormally increased levels of BNIP3 and NIX in the PPTC7 KO lines were rescued upon re-expression of wild-type PPTC7 (Fig. 1C). Collectively, our data suggest that PPTC7 is required for the turnover of BNIP3 and NIX.

The turnover of BNIP3 and NIX depends on the SCF$^{FBXL4}$ ubiquitin ligase (Cao et al, 2023; Elcocks et al, 2023; Nguyen-Dien et al, 2023). To investigate whether PPTC7 and FBXL4 operate within the same or separate pathways to regulate BNIP3 and NIX turnover, we assessed the stability of BNIP3 and NIX in CRISPR knockout cell lines lacking either PPTC7, FBXL4, or both using a cycloheximide chase assay (Fig. 1D). Our results demonstrated that the combined deficiency of both PPTC7 and FBXL4 did not lead to further stabilisation of either BNIP3 or NIX compared with the individual knockout of PPTC7 or FBXL4, suggesting that FBXL4 and PPTC7 function in a shared pathway (Fig. 1D). Strikingly, FBXL4 knockout cells displayed a notable increase of the 32 kDa (middle) form of PPTC7 (Fig. 1D). The 32 kDa form of PPTC7 also accumulated after DFP treatment (Fig. EV1C).

Immunofluorescence-based complementation assays in FBXL4- and PPTC7-deficient cells showed that the ability of either PPTC7 or FBXL4 to mediate BNIP3 and NIX turnover depends on the presence of the other protein. Rescue of either FBXL4 or PPTC7 in their respective knockout cell lines resulted in the drastic

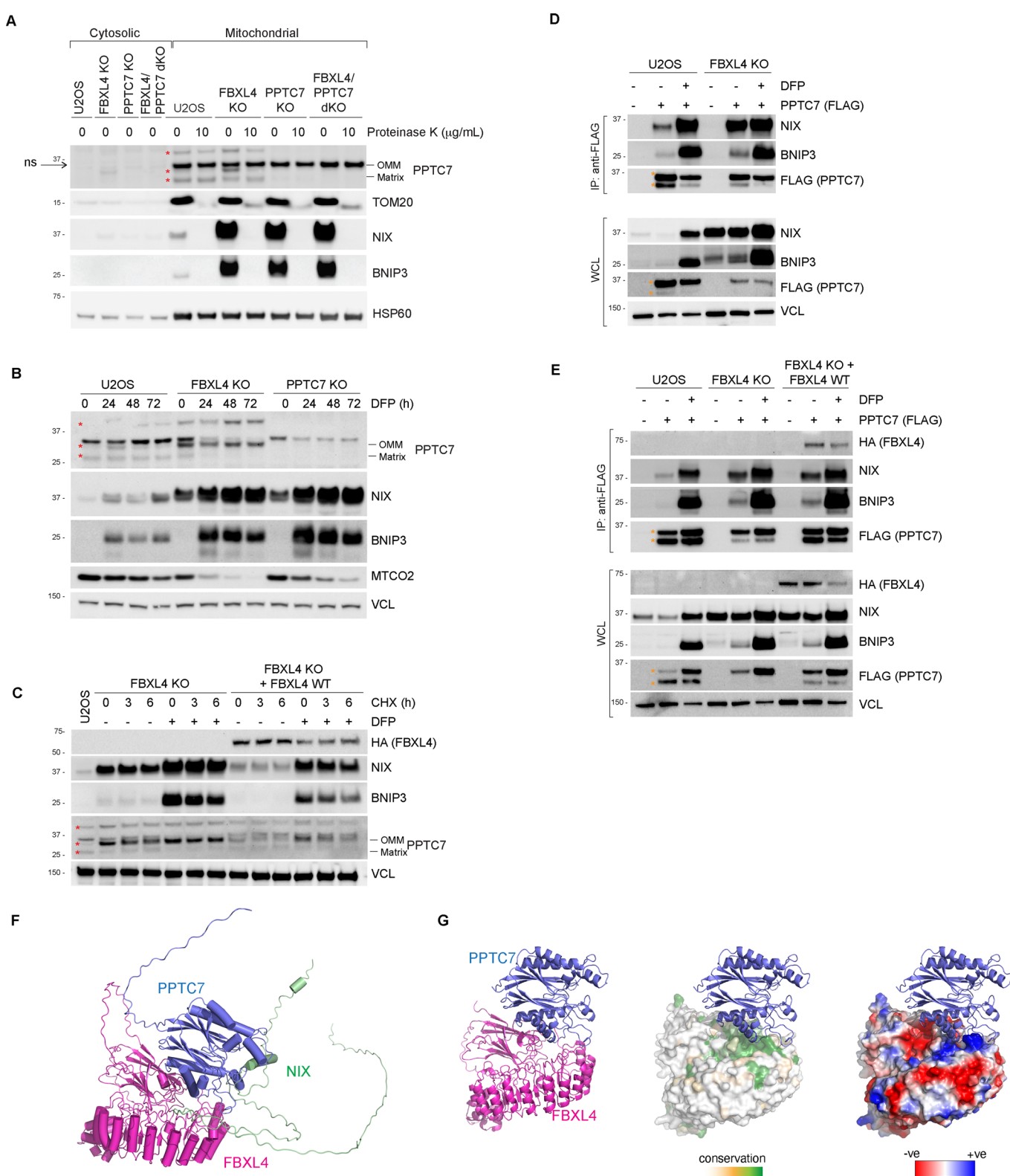

downregulation of NIX levels compared with surrounding untransfected cells (Figs. 1E,F and EV1E). In contrast, FBXL4 overexpression was not able to mediate the downregulation of BNIP3 and NIX in the absence of PPTC7 (Figs. 1E and EV1D). Likewise, PPTC7 expression could not promote BNIP3 and NIX

downregulation in the absence of FBXL4 (Figs. 1F,G and EV1E). Notably, in these experiments, PPTC7 deficiency did not affect the localisation or levels of FBXL4 (Figs. 1F and EV1D). Thus, FBXL4 and PPTC7 require each other to mediate the downregulation of BNIP3 and NIX.

◀ **Figure 2. A subset of PPTC7 localises to the mitochondria outer membrane to enable its interaction with BNIP3 and NIX and FBXL4.**

(A) The 32 kDa migrating form of PPTC7 is located on the mitochondrial outer membrane, while the lower migrating form of PPTC7 resides in the matrix. Mitochondria were isolated from parental U2OS, FBXL4 KO, PPTC7 KO or PPTC7/FBXL4 dKO cells. The submitochondrial localisation of PPTC7 was determined by proteinase K treatment, which degrades proteins that are not imported within the mitochondria, i.e., outer membrane proteins. (B) The outer membrane form of PPTC7 is induced by DFP treatment and in FBXL4-deficient cells. U2OS cells or FBXL4 KO cells were treated with DFP for the indicated times. (C) Analysis of the stability of the outer membrane form of PPTC7 in response to FBXL4 deficiency. FBXL4 KO cells and FBXL4 KO cells expressing FBXL4(HA) were subjected to cycloheximide chase assay in the presence or absence of DFP. DFP treatment and FBXL4 deficiency both extended the half-live of OM-PPTC7. (D) PPTC7 interacts with NIX/BNIP3 in basal conditions in an FBXL4-independent manner. U2OS cells or FBXL4 KO cells were transfected with PPTC7-FLAG and treated with DFP for 24 h. Cell lysates were immunoprecipitated with anti-FLAG beads, and the immuno-precipitates were analysed by immunoblotting as indicated. WCL whole-cell lysates. (E) PPTC7 interacts with FBXL4. As in (E), including PPTC7-FLAG immunoprecipitations in FBXL4 KO cells rescued with HA-tagged FBXL4 WT. (F) AlphaFold2 multimer model of FBXL4, PPTC7 and NIX. See also Fig. S2F,G. (G) The FBXL4 pocket is coloured to indicate surface sequence conservation or surface electrostatics using the Consurf or Protein-Sol Patches server. Data Information: (A–C) The red asterisks indicate PPTC7-specific bands detected by immunoblotting. The arrow points to the non-specific band (ns) at 36 kDa which is enriched in mitochondria. (D, E) The orange asterisks denote PPTC7(FLAG) expression. Source data are available online for this figure.

To determine whether PPTC7 levels are rate-limiting for BNIP3 and NIX turnover and consequently for mitophagy suppression, we examined the levels of BNIP3 and NIX in U2OS and 293T cells stably overexpressing PPTC7 (at ~100-fold for U2OS and 50-fold for 293T cells). BNIP3 and NIX levels were examined in steady-state conditions or after treatment with DFP, which is an iron chelator and HIF1α activator known to promote mitophagy via BNIP3 and NIX upregulation (Allen et al, 2013; Zhao et al, 2020). The low levels of BNIP3 and NIX in steady-state conditions made it hard to detect a further decrease in levels upon PPTC7 over-expression in U2OS, however, we found that the overexpression of wild-type PPTC7 resulted in the downregulation of BNIP3 and NIX after DFP treatment in U2OS cells and 293T cells (Fig. EV1F–H). Consistent with its suppression of BNIP3 and NIX protein levels, the overexpression of PPTC7 suppressed DFP-induced mitophagy in U2OS cells (Fig. EV1H). Furthermore, PPTC7 knockout U2OS cells treated with DFP exhibited substantially more mitophagy than either condition alone, indicating that PPTC7 suppresses both basal and DFP-induced mitophagy (Fig. 1H). Altogether, our experiments indicate that PPTC7 is rate-limiting for FBXL4-mediated BNIP3 and NIX turnover and the associated mitophagy suppression.

## A subset of PPTC7 localises to the outer mitochondrial membrane and interacts with FBXL4 and NIX/BNIP3

PPTC7 has been previously localised to the mitochondrial matrix (Guo et al, 2017; Rhee et al, 2013), whereas BNIP3 and NIX are located at the mitochondrial outer membrane. We sought to clarify the submitochondrial localisation of PPTC7 to explore whether PPTC7 is located at both the outer mitochondrial membrane as well as inside the mitochondria. Like other mitochondrial proteins, PPTC7 possesses an N-terminal mitochondrial targeting pre-sequence (MTS) that is predicted to be proteolytically cleaved by mitochondrial proteases after import into the mitochondria (Stojanovski et al, 2007), giving rise to a shorter processed form of the protein. Given the molecular weight difference predicted from the removal of the MTS, we posited that the different bands of PPTC7 might represent the precursor form of PPTC7 that has not been imported into mitochondria, and a processed shorter matrix form. To test this hypothesis, we conducted mitochondrial import assays, employing proteinase K to degrade proteins that reside on the outside of the mitochondria (i.e., that are not yet imported) (Fig. 2A). We observed that the shorter molecular weight form of PPTC7 was resistant to proteinase K suggesting that it is inside the

mitochondria. The 40 kDa (upper) form of PPTC7 was also proteinase K resistant, indicating that it likely resides in the matrix. Currently, the nature of the third 40 kDa form of PPTC7 remains unclear, however, we note that we only observe it for endogenous PPTC7 and not exogenous PPTC7.

In contrast, we found that the 32 kDa (middle) version of PPTC7, which accumulates in FBXL4-deficient cells, is susceptible to proteinase K, indicating its localisation on the mitochondrial outer membrane. Hereafter, we refer to the 32 kDa form of PPTC7 as outer membrane PPTC7 (OM-PPTC7), and the 28 kDa form of PPTC7 as inner mitochondrial PPTC7 (matrix-PPTC7).

We next explored the relative abundance of the outer/inner forms of PPTC7 in response to different mitophagy activators. In addition to FBXL4 deficiency, conditions known to upregulate BNIP3 and NIX-dependent mitophagy, such as the hypoxia-mimetics DFP and DMOG, resulted in the upregulation of the OM-PPTC7 (Figs. 2B,C and EV2A,B), largely correlating with BNIP3 and NIX levels. The greatest upregulation of OM-PPTC7 was observed in FBXL4-deficient cells, and in this condition, the levels of matrix-PPTC7 were inversely correlated with OM-PPTC7. Note that the non-specific band detected by the PPTC7 antibody disappeared in the conditions associated with the highest mitophagy, which we presume is due to a decrease in mitochondrial content caused by elevated mitophagy. In the same conditions, the decrease in matrix-PPTC7 could theoretically reflect either compromised mitochondrial import of PPTC7 or high levels of mitochondrial degradation through mitophagy or a combination of both and warrants further investigation.

Next, we used affinity purification of FLAG-tagged PPTC7 to demonstrate that PPTC7 can robustly interact with both BNIP3 and NIX (Fig. 2D,E) and FBXL4 (Fig. 2E). The interaction between PPTC7 and BNIP3/NIX was evident in basal conditions, as well as after DFP treatment. We also established that FBXL4 is not required for the interaction between PPTC7 and BNIP3/NIX using FBXL4-deficient cells (Fig. 2D,E). In our experimental conditions, it was not possible to conclusively determine whether the interactions between PPTC7 with BNIP3, NIX or FBXL4 changed in response to DFP treatment since they were confounded by the changing levels of proteins after DFP (increased BNIP3/NIX after DFP, and decreased FBXL4) and require further investigation. Lastly, we tested whether PPTC7 is required for the ability of FBXL4 to bind to SKP1 and CUL1, core members of the SCF complex. FBXL4 interacted equally with CUL1 and SKP1 in PPTC7-deficient cells, suggesting that PPTC7 is not required for SCF assembly (Fig. EV2C).

To confirm the direct interaction between PPTC7 and BNIP3/NIX we tested the binding of BNIP3 and NIX synthetic peptide sequences to recombinant PPTC7 by isothermal titration calorimetry (ITC). These peptides were based on the structural modelling described below. We first tested the functionality of PPTC7 by measuring the association of the PPTC7 active site with divalent cations, observing binding of both $Mg^{2+}$ and $Mn^{2+}$, but with a much higher affinity for $Mn^{2+}$ (Fig. EV2D; Appendix Table S1) in line with the previous demonstration of $Mn^{2+}$-dependent activity (Guo et al, 2017). Subsequently, we found that both BNIP3 and NIX peptides bound to PPTC7 directly, with similar although modest affinities ($K_d$) of 20 and 37 μM, respectively (Fig. EV2E; Appendix Table S1).

Since BNIP3 and NIX demonstrated binding to PPTC7, but little to no binding with FBXL4 in our experimental conditions (Fig. EV2B), this led us to speculate that PPTC7 might serve as a bridge or scaffold between FBXL4 and BNIP3/NIX. To examine this at the molecular level, we modelled the interactions between PPTC7, FBXL4 and BNIP3/NIX using AlphaFold2 (Figs. 2F and EV2F,G). Predictions of pairwise complexes or all three proteins together resulted in identical structural models. Alphafold2 predicts a high-confidence interaction between FBXL4 and one surface of PPTC7, while a conserved sequence found in both BNIP3 and NIX associates with the active site of PPTC7. A conserved and negatively charged pocket formed between FBXL4's N-terminal discoidin domain and C-terminal LRR domains surrounds PPTC7's residues E125-K130 (Figs. 2F,G and EV2F,G). The binding of BNIP3 and NIX involves a highly extended peptide sequence of ~25 residues including the distal end of BNIP3/NIX's non-canonical BH3 domains, centred on a highly conserved sequence, which we hereafter refer to as the SRPE sequence (encompassing residues 122–125 and 146–149 and in BNIP3 and NIX, respectively). The conserved Ser sidechain is predicted to bind the catalytic pocket, precisely where a phosphorylated substrate would be expected to interact. These structural predictions are consistent with PPTC7 serving to position BNIP3/NIX to be substrates of the SCF^FBXL4 E3 complex.

## Disruption of the catalytic activity of PPTC7 does not affect BNIP3 and NIX turnover in basal conditions

BNIP3 and NIX are known to be phosphoproteins, and PPTC7 knockout systems have shown elevated phosphorylation at specific residues on BNIP3 and NIX (Niemi et al, 2023), albeit not within the regions of BNIP3 and NIX predicted to interact with PPTC7. Specifically, interrogation of existing phosphorylation databases finds no evidence of phosphorylation on NIX's Ser146 or BNIP3's Ser122, which are situated within the PPTC7 active site.

To determine whether the phosphatase activity of PPTC7 is required for BNIP3 and NIX turnover, we engineered mutant versions of PPTC7 predicted to have disrupted phosphatase activity. This involved mutating the active site residues, which were identified through a comparison of the AlphaFold2 prediction of PPTC7 (ID: AF-Q8NI37-F1) with the crystal structure of the similar PPM/PP2C family homologue photosystem II (PSII) core phosphatase (PBCP) (PDB ID: 6AE9) (Liu et al, 2019). Specifically, we substituted the metal-binding aspartate residues at positions Asp78, Asp233 and Asp290 in PPTC7 (Figs. 3A and EV3A). We found that PPTC7 mutants in which Asp78 and Asp223 were

mutated to alanine (PPTC7-D78A and PPTC7-D223A) were unable to rescue BNIP3 and NIX turnover when complemented into PPTC7-deficient cells (Fig. EV3B,C). However, these mutations also significantly reduced the binding of PPTC7 to BNIP3 and NIX (Fig. EV3D), either due to the loss of metal ion-binding required to coordinate the Ser sidechain in the SRPE BNIP3/NIX binding sequences, or possibly due to conformational rearrangements. Thus, it was not possible using these mutations to ascertain if the lack of turnover was due to a lack of phosphatase activity or reduced binding to BNIP3/NIX.

To preserve the metal-binding activity of PPTC7 and thus the interaction between PPTC7 and BNIP3/NIX, we next substituted the polar aspartate residues with alternate polar asparagine residues (D78N, D223N or D290N). This change preserved partial binding between PPTC7 and BNIP3/NIX, with the PPTC7-D290N variant displaying the greatest binding compared with wild-type PPTC7 (Fig. 3B).

To determine if the catalytic activity of PPTC7 is important for BNIP3 and NIX degradation, we expressed PPTC7-wildtype or PPTC7-aspartate to asparagine active site mutants in PPTC7 KO cells. Complementation assays demonstrated that PPTC7-D290N can fully rescue the turnover of BNIP3 and NIX (Fig. 3C,D), with partial rescue observed for PPTC7-D78N and PPTC7-D223N. We confirmed that PPTC7-D290N had compromised catalytic activity compared with PPTC7-wildtype by assessing the BNIP3 phospho-migration shift, finding purified wild-type PPTC7 could dephosphorylate BNIP3 however PPTC7-D290N could not (Fig. 3E). PPTC7-D290N was also confirmed to be catalytically defective using pNPP phosphatase assays (Fig. 3F). Our data demonstrate that although BNIP3 (and likely NIX) can be dephosphorylated by PPTC7 (Fig. 3E), the FBXL4-mediated degradation of BNIP3 and NIX is still mediated by the catalytically defective PPTC7, suggesting that the turnover of PPTC7 depends on its presence, but not its full activity.

## BNIP3/NIX–PPTC7 interactions are critical for BNIP3/NIX turnover and mitophagy suppression

To further test the significance of the interaction between NIX/BNIP3 and PPTC7, we sought to disrupt the interaction between BNIP3/NIX and PPTC7. Based on our models of the PPTC7 complex with either BNIP3 or NIX using Alphafold2 (Bryant et al, 2022), Tyr179 and Asn181 in PPTC7 were predicted to be critical residues required for binding of BNIP3/NIX without affecting the PPTC7 active site (Fig. 4A,D; S4A). On the BNIP3/NIX interface, Trp144 in the BH3 domain, and a highly conserved SRPE region in NIX/BNIP3 were predicted to interact with PPTC7's active site (residues 122–125 in BNIP3 and 146–149 and NIX).

To test the Alphafold2 prediction, we made the following mutations in PPTC7: Y179D to disrupt the interface with BNIP3/NIX and N181E to disrupt the local pocket around the phosphoserine-binding catalytic site (Fig. 4A). First, we tested whether these mutants lose binding to BNIP3/NIX using an anti-FLAG affinity pulldown assay. Our results demonstrated that PPTC7-Y179D and PPTC7-N181E have reduced binding to both BNIP3 and NIX in co-immunoprecipitation experiments (Fig. 4B), indicating that Tyr179 and Asn181 residues are important for the interaction with BNIP3/NIX. In contrast, Gln128 and Lys130 lie in the putative FBXL4 binding site, and Q128R and K130E mutations in PPTC7 did not affect BNIP3/NIX binding, as predicted (Fig. 4B),

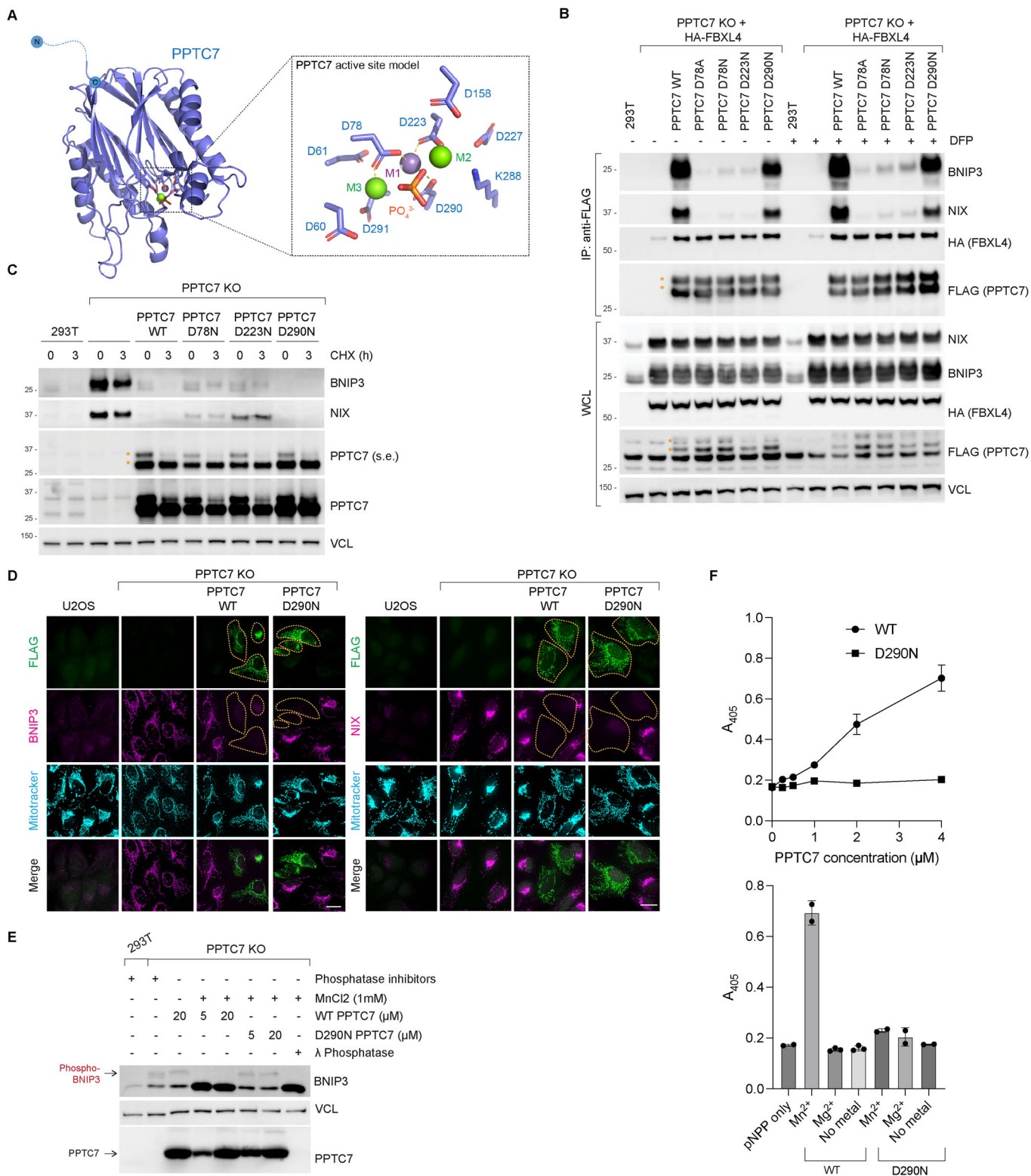

acting as controls for testing BNIP3/NIXinteractions. We confirmed that the Y179D mutant had reduced binding to NIX using isothermal titration calorimetry (ITC) experiments (EV4B).

Secondly, we tested whether the disruption of PPTC7's ability to interact with BNIP3/NIX disrupted the ability of PPTC7 to mediate

BNIP3/NIX turnover. As hypothesised, the loss of PPTC7's interaction with BNIP3 and NIX resulted in a loss of its ability to mediate BNIP3 and NIX turnover (Fig. 4C). We performed rescue assays in PPTC7 KO cell lines using either wild-type PPTC7 or the variants in PPTC7 with defective binding to BNIP3/NIX: PPTC7-

◄ **Figure 3.  Disruption of the catalytic activity of PPTC7 does not affect BNIP3 and NIX turnover in basal conditions.**

(A) AlphaFold2 model of PPTC7 with active site residues indicated. This is based on a comparison to the PBCP structure (PDB ID 6AE9). The three putative metal ions and the incoming phosphate group are modelled based on an alignment of the PPTC7 and PBCP structures. (B) Binding to BNIP3 and NIX is preserved in the PPTC7-D290N mutants. PPTC7 KO 293T cells expressing FBXL4(HA) were transfected with PPTC7-FLAG or D78N, D223N, and D290N mutants. Cells were treated with DFP for 24 h where indicated. Cell lysates were immunoprecipitated with anti-FLAG beads, and the immuno-precipitates were analysed by immunoblotting as shown. (C) PPTC7-D290N can restore the turnover of BNIP3 and NIX to a similar extent as wild-type PPTC7. The PPTC7 KO cells were rescued with wild-type PPTC7-FLAG or PPTC7-D78N, PTPC7-D223N or PPTC7-D290N variants. Cells were treated with cycloheximide for 3 h before harvesting. Samples were lyzed, and immunoblotting was performed. s.e. short exposure. (D) PPTC7-D290N can rescue the degradation of BNIP3 and NIX to a similar extent as wild-type PPTC7. NIX and BNIP3 protein levels (magenta) were monitored in PPTC7(FLAG) expressing cells. To correlate BNIP3/NIX levels with PPTC7 expression, PPTC7 expressing cells (green) are outlined in orange dotted lines. Scale bar = 20 microns. (E) PPTC7 wild-type can dephosphorylate BNIP3 but PPTC7-D290N cannot. Electrophoretic mobility shift assays of BNIP3 using phos-tag gels were performed to assess the phosphorylation status of BNIP3. Purified PPTC7 or PPTC7-D290N were incubated with 293T PPTC7 KO cell lysates. Wild-type PPTC7 could dephosphorylate BNIP3 in a $Mn2+$-dependent manner as demonstrated by the loss of the upper form of BNIP3 (arrow). However, PPTC7-D290N was not able to dephosphorylate BNIP3 in the same conditions. Lamda-phosphatase is used as a control to demonstrate that the upper band of BNIP3 seen on the phos-tag gels is the phosphorylated species. (F) In vitro pNPP dephosphorylation assay using purified PPTC7 wild-type and PPTC7-D290N. Top Panel: The dephosphorylation of pNPP was measured at 405 nm after a 15-min reaction using increasing concentrations of PPTC7. Bottom panel: 10 mM pNPP was added to 4 µM PPTC7 in the presence or absence of either $MnCl_2$ or $MgCl_2$. Bars represent mean values $+/-$ standard deviation. $n = 1$, individual data points represent technical replicates. Data Information: (B, C) The orange asterisks denote PPTC7(FLAG) expression. Source data are available online for this figure.

Y179D and PPTC7-N181E. We found that, unlike wild-type PPTC7 which reduced the elevated BNIP3 and NIX levels, PPTC7-Y179D and PPTC7-N181E displayed elevated BNIP3/NIX compared with wild-type PTC7 as assessed using a cycloheximide chase assay (Fig. 4C). The mutants localised to the mitochondria as expected (Fig. EV4C).

To functionally validate the regions on BNIP3 and NIX required for their interaction with PPTC7, a series of NIX or BNIP3 mutation constructs were expressed in BNIP3/NIX double knock-out cells: NIX-W144D, NIX-Δ140–150, NIX-R147D, and triple substitutions of R-P-E$_{(147-149)}$ to either alanine (RPE-AAA) or aspartic acid and alanines (RPE-DAA). We performed FLAG affinity purification assays in HeLa Flp-In BNIP3/NIX double KO cells co-expressing doxycycline-inducible FLAG-tagged wild-type NIX or NIX mutants along with HA-tagged PPTC7 (Figs. 4E and EV4D). Indeed, we found that despite the lower levels of wild-type NIX, only wild-type NIX was able to bind to PPTC7 (Figs. 4E and EV4D), whereas NIX-W144D, NIX-RPE/AAA, NIX-RPE/DAA and NIX--Δ140–150 were not. Fitting with the hypothesis that the interaction occurs at the outer membrane, the OM-form of PPTC7 bound preferentially to NIX. The expression of PPTC7-WT promoted the downregulation of wild-type NIX, but not the PPTC7-binding NIX variants, supporting the model that the binding of PPTC7 to NIX is required for its degradation. Therefore, the region surrounding the SRPE domain in NIX is required for functional interaction with PPTC7.

Having demonstrated that NIX's SRPE and nearby residues are important for PPTC7 binding, we next assessed their stability. Supporting the Alphafold2 model, each of the NIX mutants displayed higher expression than wild-type NIX (Figs. 4F and EV4E). These results suggest that the PPTC7-binding region in NIX plays an important role in its turnover.

Since elevated levels of BNIP3 and NIX correlate with increased mitophagy, we tested the hypothesis that the expression of the PPTC7-resistant hyper-stable NIX mutants would result in elevated mitophagy using a mt-Keima assay (Figs. 4G and EV4F). In our conditions, the expression of wild-type NIX does not induce mitophagy since it is expressed at low levels, enabling a direct comparison of how stabilising NIX by preventing its interaction with PPTC7 influences mitophagy. We expressed NIX-RPE/AAA, NIX-RPE/DAA and NIX-W144D in mt-Keima-expressing and

compared the induction of mitophagy to that induced by wild-type NIX (Figs. 4G and EV4F). We found that the expression of NIX-RPE/AAA and RPE/DAA resulted in substantially elevated mitophagy in cells compared with wild-type NIX, indicating that the stabilisation of NIX due to loss of its binding to PPTC7 results in hyperactivation of mitophagy.

Taken together, these results validate the Alphafold2 modelling and demonstrate the importance of the BH3-SRPE region of NIX and the Tyr179 and Asn181 residues of PPTC7 in NIX-PPTC7 binding. Furthermore, the results demonstrate that PPTC7 interaction is important for BNIP3/NIX turnover and mitophagy suppression.

## The FBXL4-PPTC7 interaction is critical for mitophagy receptor turnover and mitophagy suppression

To test whether PPTC7's interaction with FBXL4 is required for BNIP3 and NIX turnover, we proceeded to explore PPTC7's interaction with FBXL4 using AlphaFold2 modelling. The predictions indicated that FBXL4's Met71 and Arg544 are important for interaction with PPTC7 (Fig. 5A), therefore we generated the following FBXL4 mutants to test binding to PPTC7: FBXL4-M71E, FBXL4-R544E or FBXL4-M71E/R544E. Consistent with the structural models, we found that FBXL4 mutants displayed weaker binding to PPTC7 than wild-type FBXL4, with the greatest reduction in binding to PPTC7 observed for FBXL4-M71E/R544E (Fig. 5B). The binding of BNIP3 and NIX to PTTC7 was greatly increased in cells expressing the FBXL4 mutants that were unable to bind to PPTC7 (FBXL4-M71E, M544E and M71E/R544E). This increase in binding is likely due to their increased levels, rather than any increase in affinity although our data does not rule this out. However, the result confirms that the interaction between FBXL4 and PPTC7 is not required for the interaction between PPTC7 and BNIP3/NIX (Fig. 5B).

We next assessed the functional significance of this interface by assessing whether the loss of the binding between FBXL4 and PPTC7 affects the ability of FBXL4 to mediate the turnover of BNIP3 and NIX. Rescue assays were performed in FBXL4 KO cells expressing FBXL4 wild-type, FBXL4-M71E, FBXL4-E544E or FBXL4-M71E/R544E and the stability of BNIP3 and NIX were assessed using a cycloheximide chase assay (Fig. 5C). We found

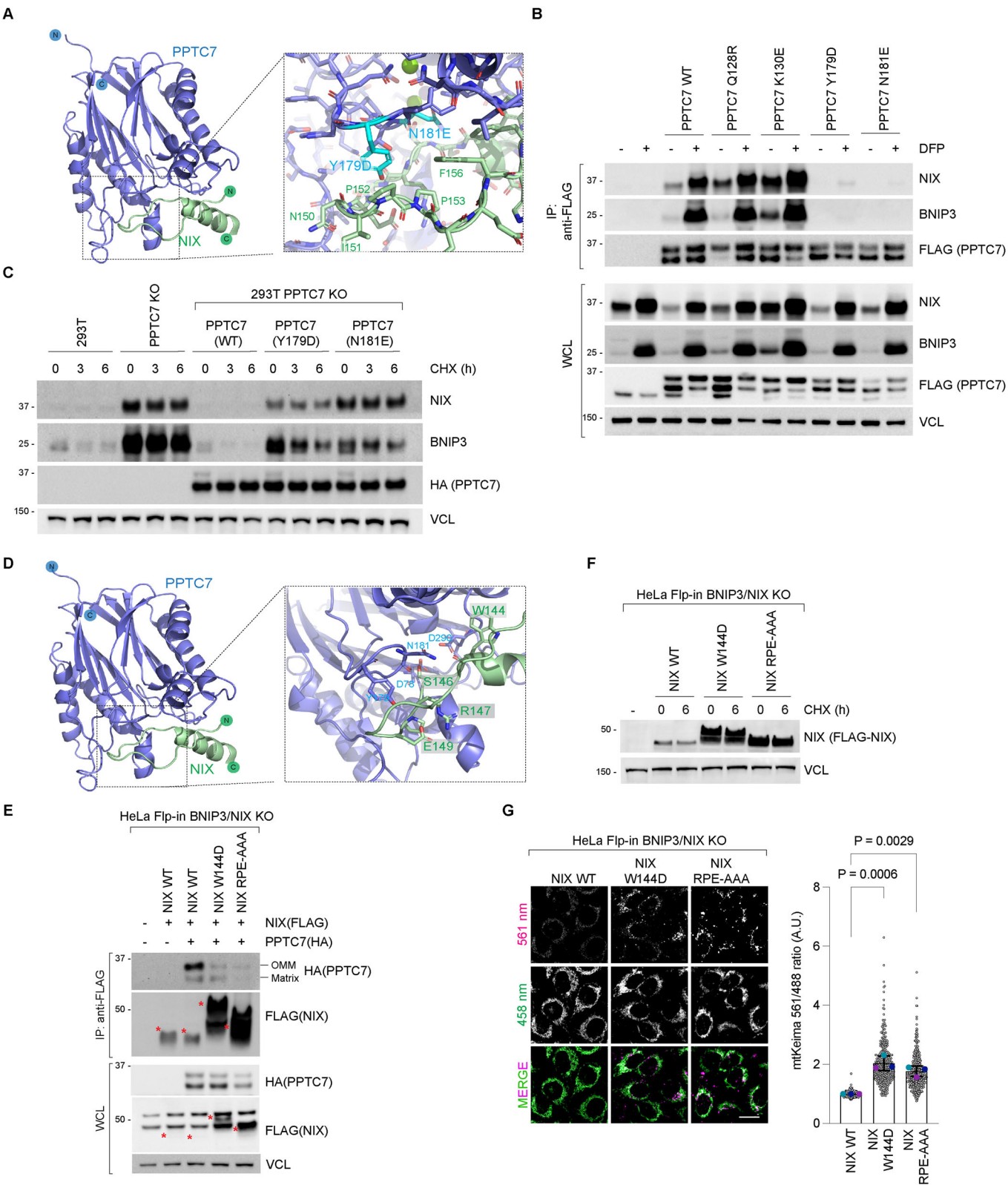

◀ **Figure 4. The NIX-PPTC7 interaction is critical for NIX turnover and mitophagy suppression.**

(A) AlphaFold2 prediction of the PPTC7–NIX complex. Key residues in the PPTC7 interface are highlighted in cyan. (B) PPTC7-Y179D and PPTC7-N181E variants are unable to interact with BNIP3 and NIX. PPTC7(FLAG) and specified mutants were transfected into U2OS cells. Cells were treated with DFP for 24 h. Cell lysates were precipitated with anti-FLAG affinity resin and the immuno-precipitates were analysed by immunoblotting. (C) PPTC7-Y179D and PPTC7-N181E variants are unable to mediate the turnover of BNIP3 and NIX. PPTC7(HA)-WT, PPTC7-Y179D or PPTC7-N181E were expressed in PPTC7 KO 293T cells. BNIP3 and NIX stability was assessed using a cycloheximide chase. (D) AlphaFold2 prediction of the PPTC7–NIX complex. Key residues in the NIX interface including Arg147 are highlighted. (E) Residues W144 and RPE$_{147-149}$ in NIX are critical for binding to OM-PPTC7. PPTC7(HA) was transduced into HeLa Flp-in BNIP3/NIX double KO cells stably expressing inducible FLAG-tagged NIX-WT or NIX mutants. NIX expression was induced with doxycycline for 24 h. Cell lysates were immunoprecipitated with anti-FLAG beads, and the immuno-precipitates were analysed by immunoblotting. The levels of NIX-W144D and NIX-RPE/AAA were significantly higher than NIX-wild-type. Red asterisks mark the NIX(FLAG) proteins. The different migration of the point mutants by electrophoresis could be due to the change in charge of the residues or because PPTC7 is unable to dephosphorylate NIX. (F) NIX-W144D and NIX-RPE/AAA are expressed at higher levels in comparison to wild-type NIX. HeLa Flp-In BNIP3/NIX double knockout Keima cells stably expressing NIX and NIX mutants were treated with doxycycline for 48 h. Cells were subjected to a cycloheximide chase. (G) Expression of hyper-stable NIX-W144D and NIX-RPE/AAA leads to an increase in steady-state mitophagy compared with NIX-wild-type. HeLa Flp-In BNIP3/NIX double knockout Keima cells stably expressing NIX and NIX mutants were treated with doxycycline for 48 h and mitophagy was evaluated using live-cell confocal fluorescence microscopy. Mitophagy is represented as the ratio of mt-Keima 561 nm fluorescence intensity divided by mt-Keima 458 nm fluorescence intensity for individual cells normalised to mitophagy observed in cells expressing NIX-wild-type. Translucent grey dots represent measurements from individual cells. Coloured circles represent the mean ratio from independent experiments. The centre lines and bars represent the mean of the independent replicates $+/-$ standard deviation. P values were calculated based on the mean values using a one-way ANOVA. Source data are available online for this figure.

that the double mutant of FBXL4-M71E/R544E was unable to reduce the stability of BNIP3 and NIX to the same level as FBXL4 wild-type. Similar to FBXL4 deficiency, the outer membrane form of PPTC7 was also stabilised in cells expressing FBXL4 variants unable to bind to PPTC7. Notably, despite their reduced ability to downregulate BNIP3 and NIX (Fig. 5B,C), the FBXL4 variants localised, like wild-type FBXL4, to mitochondria (Fig. 5D).

Finally, we validated that these residues are important for the ability of FBXL4 to suppress mitophagy, finding that FBXL4-M71E/R544E was also less effective at suppressing mitophagy when reconstituted into FBXL4-deficient cells, correlating with increased BNIP3 and NIX levels in cells expressing the variants compared with wild-type FBXL4 (Fig. 5E). In all, these results suggest that the interaction with PPTC7 is required for FBXL4 to regulate BNIP3/NIX turnover and for the ability of FBXL4 to suppress mitophagy.

## Discussion

In this study, we provide evidence that PPTC7 facilitates the SCF$^{FBXL4}$-mediated turnover of BNIP3 and NIX. Our findings suggest that PPTC7 is a critical rate-limiting factor determining the amount of BNIP3 and NIX turnover and that it operates directly at the outer mitochondrial membrane by interacting with BNIP3/NIX as well as with FBXL4, rather than employing an indirect mechanism.

PPTC7 accumulates at the mitochondrial outer membrane in response to pseudohypoxia and other conditions associated with elevated BNIP3/NIX-mediated mitophagy. The observed increase in outer membrane PPTC7 in response to high mitophagy suggests a potential homoeostatic feedback mechanism to limit excessive mitophagy. However, it remains incompletely understood how PPTC7's localisation to the outer membrane is regulated. PPTC7 may be co-regulated with BNIP3 and NIX by FBXL4-mediated turnover. Alternatively, homoeostasis may be achieved by modulating the rate of PPTC7 import into mitochondria. The accumulation of outer membrane PPTC7 may be a consequence of defective protein import, active retention mechanisms, or a combination of these possibilities. It also remains an open question whether this regulation might occur in a localised manner, perhaps targeting specific mitochondria (Fig. 6A).

While we were preparing this manuscript, a study by Sun and colleagues was published, which contains highly complementary findings to ours about the post-transcriptional regulation of BNIP3 and NIX through the FBXL4 and PPTC7 (Sun et al, 2024). Their findings, akin to ours, show that PPTC7 acts as a critical and limiting factor in governing the FBXL4-mediated degradation of BNIP3 and NIX. Additionally, both studies highlight the dual localisation of PPTC7 to the matrix as well as the mitochondria outer membrane to enable its interaction with BNIP3 and NIX. Similarly, using different approaches to inhibit PPTC7, Sun et al suggest that the full catalytic activity of PPTC7 is not required for the turnover of BNIP3 and NIX, paralleling our observations. Sun et al additionally present elegant physiological data suggesting that the upregulation of PPTC7 in the context of the liver during fasting conditions is required to maintain mitochondrial numbers in this organ. An interesting area needing further clarification in the future is the working model for how PPTC7 scaffolds the essential interactions between FBXL4 and BNIP3/NIX and/or the SCF complex itself. In this regard, our data diverges from the working model of Sun et al, suggesting that rather than promoting SCF complex assembly through CUL1 recruitment, PPTC7 acts as an adaptor between FBXL4 and BNIP3. Further investigation incorporating structural and mechanistic data is required to reconcile these interesting findings. Altogether, the considerable overlap between our findings and those of Sun and colleagues underscores the indispensable role of the PPTC7-FBXL4 axis in suppressing mitophagy.

Our findings show that the upregulation of BNIP3 and NIX due to PPTC7 or FBXL4 disruption leads to mitophagy in only a subset of mitochondria. This suggests that while the increased expression of BNIP3 and NIX is necessary to trigger mitophagy, additional factors are required for its full induction, or that mitochondrial biogenesis compensates for the increased mitophagy.

In silico modelling suggests that PPTC7 bridges the interaction of BNIP3 and NIX with FBXL4, enabling the effective positioning of the SCF complex for productive ubiquitylation of BNIP3/ NIX. This conclusion is further supported by functional assays, which demonstrate that disrupting the interactions between NIX and PPTC7, as well as between FBXL4 and PPTC7, results in the stabilisation of BNIP3 and NIX, leading to increased basal

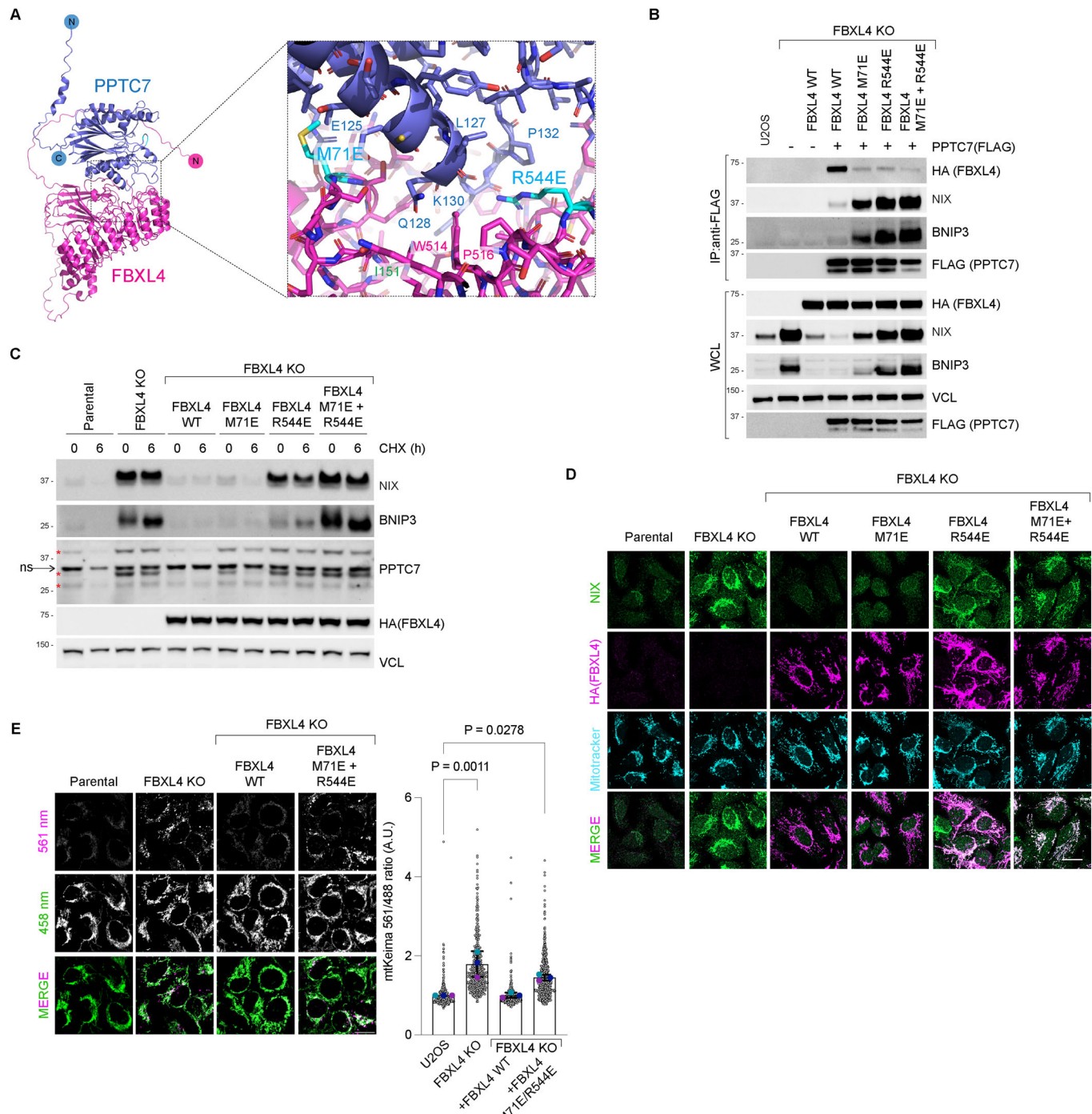

**Figure 5.  The FBXL4-PPTC7 interaction is required for BNIP3 and NIX turnover and mitophagy suppression.**

(**A**) Alphafold2 structural modelling of FBXL4 in complex with PPTC7. Alphafold2 predicts a high-confidence interaction between FBXL4 and PPTC7 centred on Met71 and Arg544 in FBXL4. (**B**) Met71 and Arg544 in FBXL4 are involved in the interaction with PPTC7. FBXL4 knockout cells were complemented with HA-tagged wild-type FBXL4, FBXL4-M71E, FBXL4-R544E or FBXL4-M71E/R544E. Cells were transfected with FLAG-tagged PPTC7 as indicated, lysed, and subjected to affinity purification using anti-FLAG resin. WCL whole-cell lysates. (**C**) FBXL4-M71E, FBXL4-R544E or FBXL4-M71E/R544E variants are unable to mediate BNIP3 and NIX downregulation and destabilisation. U2OS FBXL4 KO cells were rescued with wild-type FBXL4(HA) or specified variants. (**D**) Localisation of FBXL4-M71E, FBXL4-R544E or FBXL4-M71E/R544E variants. U2OS FBXL4 KO cells expressing FBXL4(HA) wild-type or specified variants were fixed and stained for HA to detect FBXL4 (in magenta) or NIX (green). NIX levels are inversely correlated with the ability of FBXL4 to bind to PPTC7. Scale bar = 20 microns. (**E**) FBXL4-M71E/R544E is less efficient than wild-type FBXL4 in mediating mitophagy suppression. U2OS mt-Keima cells, U2OS mt-Keima FBXL4 KO cells and FBXL4 KO cells rescued with FBXL4 constructs were analysed by confocal microscopy to quantify mitophagy. *P* values were calculated based on the mean values using a one-way ANOVA. 4 individual data points outside axis limits. *n* = 3 independent replicates. Data Information: (**C**) Red asterisks mark the PPTC7-specific bands detected by immunoblotting. Source data are available online for this figure.

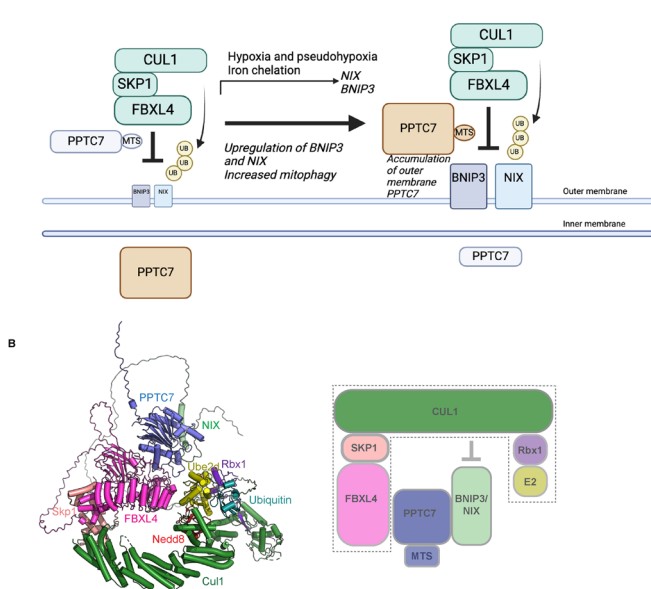

**Figure 6. Working model for FBXL4 and PPTC7 mediated turnover of BNIP3 and NIX for mitophagy suppression.**

(A) Working model for the role of PPTC7 in mitophagy suppression. In steady-state conditions, low levels of PPTC7 localise at the mitochondrial outer membrane to mediate the constitutive turnover of BNIP3/NIX, and the majority of PPTC7 localises in the mitochondrial matrix. PPTC7 accumulates at the mitochondria outer membrane in conditions of high mitophagy such as pseudohypoxia to dampen mitophagy. PPTC7 accumulation on the outer membrane may be a result of active retention mechanisms or defective mitochondrial import. (B) Model of the combination of Alphafold2 PPTC7-BNIP3-FBXL4 with the structure of Skp1-Cul1-Ube2d-Ub-Nedd8 (Baek, 2020, PDB ID 6TTU). PPTC7 interacts with BNIP3/NIX and with FBXL4. One interpretation of our data is that PPTC7 bridges the interaction between FBXL4 and BNIP3/NIX to position BNIP3/NIX substrates for productive ubiquitylation by SCF$^{FBXL4}$.

mitophagy. Our results suggest that PPTC7 binds to directly to substrates BNIP3 and NIX, rather than the SCF components. The PPTC7 scaffold function we propose resembles the function of the CKS1 accessory factor, which plays a critical role in facilitating the interaction between FBXL1 (also known as SKP2) and its substrate, the cyclin-dependent kinase (Cdk) inhibitor p27 (Hao et al, 2005) (Fig. 6B).

We note that we have not yet investigated whether certain conditions modulate FBXL4-PPTC7- BNIP3/NIX interactions. These could be local signalling events, like oxidative stress, or global conditions like starvation. Moreover, although our structure-function analyses support the significance of individual structural interfaces, it remains possible that PPTC7 may interact in a mutually exclusive manner with either NIX/BNIP3 or FBXL4 since we have not provided experimental evidence for the existence of a trimeric complex. Intriguingly, a complementary study from Wei et al recently demonstrated that PPTC7 is dynamically recruited to BNIP3 during the resolution of pseudohypoxia-induced mitophagy (preprint: (Wei et al, 2024)).

Future research should address the importance of PPTC7's catalytic activity in BNIP3 and NIX degradation. Phosphorylation of BNIP3 and NIX has been shown to influence their stability and/

or their capacity to promote mitophagy (He et al, 2022; Poole et al, 2021; Rogov et al, 2017) (Marinkovic et al, 2021; Yuan et al, 2017), suggesting that dephosphorylation may be required for their turnover and/or mitophagy suppression. Although our data suggest that PPTC7's catalytic activity is largely dispensable for BNIP3 and NIX degradation, it remains possible that small amounts of catalytic activity from the PPTC7-D290N mutant are sufficient for turnover of BNIP3 and NIX in basal conditions. Alternatively, the catalytic activity of PPTC7 may be more important in certain conditions with elevated BNIP3 and NIX, such as during hypoxia or iron chelation. PPTC7-mediated dephosphorylation may promote critical interactions required for FBXL4 to mediate BNIP3 and NIX turnover. For instance, another F-box protein, FBXL2, interacts with the PTPL1 phosphatase, which dephosphorylates p85β on Tyr-655, thereby promoting p85β binding to FBXL2 and subsequent degradation (Kuchay et al, 2013). Both BNIP3 and NIX contain a serine residue (Serine 146 in NIX and Serine 122 in BNIP3) that directly interacts with the active site of PPTC7, where a phosphorylated substrate would typically bind. However, it is important to note that we have not yet found evidence for phosphorylation of this serine residue in existing literature or our phosphor-proteomic studies. Lastly, whether the catalytic activity or levels of PPTC7 and/or SCF$^{FBXL4}$are regulated in response to environmental conditions remains a topic for further exploration.

On the other hand, if the enzymatic activity of PPTC7 is indeed dispensable for its regulation of NIX and BNIP3 turnover, this raises the question of the actual role of its catalytic activity. It is possible that PPTC7-dependent dephosphorylation of BNIP3 and NIX, or FBXL4—or perhaps regulatory proteins such as mitochondrial import proteins (Niemi et al, 2019)—could serve a distinct role on the outer membrane to suppress mitophagy. Gaining insights into the kinases that counteract PPTC7's function will be crucial for understanding how it contributes to the homoeostatic regulation of mitophagy.

## Methods

**Reagents and tools table**

| Reagent/resource | Reference or source | Identifier or catalog number |
|---|---|---|
| **Bacterial strains** | | |
| *E. coli* DH5α | Invitrogen | 18265017 |
| *E. coli* BL21(DE3) | Merck Australia | CMC0016 |
| **Antibodies** | | |
| BNIP3 (ANa-40) mouse monoclonal | Santa Cruz Biotechnology | sc-56167 |
| BNIP3 rabbit monoclonal | Abcam | ab109362 |
| Cullin 1 rabbit | Invitrogen | 718700 |
| FLAG monoclonal | Sigma-Aldrich | F3165 |
| FLAG rabbit polyclonal | Sigma-Aldrich | SAB4301135 |
| GFP | Invitrogen | MA5-15256 |
| HA rabbit monoclonal | Cell Signalling Technology | 3724S |
| HDAC6 | Santa Cruz Biotechnology | sc-11420 |
| HIF1α rabbit monoclonal | Cell Signalling Technology | 36169S |

| Reagent/resource | Reference or source | Identifier or catalog number |
|---|---|---|
| MTCO2 mouse monoclonal | Abcam | ab110258 |
| MYC | Bethyl Technologies | A190-105A |
| NIX mouse monoclonal | Santa Cruz Technologies | sc-166332 |
| NIX rabbit polyclonal | Cell Signalling Technology | 12396 |
| p27 mouse monoclonal | BD Biosciences | 610242 |
| PPTC7 rabbit | Novus Biologicals | NBP1-90654 |
| Skp1 | Pagano Laboratory | |
| TOMM20 mouse monoclonal | BD Biosciences | Clone 29 612278 |
| HSP60 | | |
| Vinculin (VCL) mouse monoclonal | Santa Cruz Biotechnology | sc-55465 |
| Donkey anti-mouse IgG Alexa Fluor™ 488 | Invitrogen | A21202 |
| Donkey anti-mouse IgG Alexa Fluor™ 555 | Invitrogen | A31570 |
| Donkey anti-mouse IgG Alexa Fluor TM 647 | Invitrogen | A31571 |
| Donkey anti-rabbit IgG Alexa Fluor™ 488 | Invitrogen | A21026 |
| Donkey anti-rabbit IgG Alexa Fluor™ 555 | Invitrogen | A31572 |
| Donkey anti-rabbit IgG Alexa Fluor™ 647 | Invitrogen | A31573 |
| **Chemicals** | | |
| Benzamidine hydrochloride hydrate | Sigma-Aldrich | B6506 |
| Deoxyribonuclease I (DNase I) | Sigma-Aldrich | DN25 |
| Talon® resin | Clontech | 635503 |
| Glutathione Sepharose 4B | GE Healthcare | GEHE17-0756-0 |
| Isopropyl β-D-1-thiogalactopyranoside | Bioline | BIO-37036 |
| **Recombinant DNA** | | |
| PPTC7-1_pLentiCRISPR v2 | This paper | |
| PPTC7-2_pLentiCRISPR v2 | This paper | |
| Fbxl4 CRISPR guide RNA 2_pSpCas9 BB-2A-Puro (PX459) v2.0 | Nguyen-Dien | |
| hPPTC7_pcDNA3.1(+)-C-DYK | This paper | |
| hPPTC7-pcDNA3.1-FLAG | Natalie Niemi | |
| pLV[Exp]-Puro-EF1A>hPPTC7[NM_139283.2]/HA | This paper | |
| FBXL4-pcDNA3.1-3XFLAG | This paper | |
| D290N_hPPTC7_pcDNA3.1(+)-C-DYK | This paper | |
| D223N_hPPTC7_pcDNA3.1(+)-C-DYK | This paper | |
| D78N_hPPTC7_pcDNA3.1(+)-C-DYK | This paper | |
| pLV_Puro_EF1A_PPTC7_FLAG | This paper | |
| D78N_pLV_Puro_EF1A_PPTC7_FLAG | This paper | |
| D223N_pLV_Puro_EF1A_PPTC7_FLAG | This paper | |
| D290N_pLV_Puro_EF1A_PPTC7_FLAG | This paper | |
| D78A_hPPTC7_pcDNA3.1(+)-C-DYK | This paper | |
| D78A/D223A_hPPTC7_pcDNA3.1(+)-C-DYK | This paper | |
| D223A_D78A_pLV_Puro_EF1A_PPTC7_HA | This paper | |
| D78A_pLV_Puro_EF1A_PPTC7_HA | This paper | |
| K130E_hPPTC7_pcDNA3.1(+)-C-DYK | This paper | |
| Q128R_hPPTC7_pcDNA3.1(+)-C-DYK | This paper | |
| Y179D_hPPTC7_pcDNA3.1(+)-C-DYK | This paper | |

| Reagent/resource | Reference or source | Identifier or catalog number |
|---|---|---|
| N181E_hPPTC7_pcDNA3.1(+)-C-DYK | This paper | |
| Q128R_pLV_Puro_EF1A_PPTC7_HA | This paper | |
| K130E_pLV_Puro_EF1A_PPTC7_HA | This paper | |
| Y179D_pLV_Puro_EF1A_PPTC7_HA | This paper | |
| N181E_pLV_Puro_EF1A_PPTC7_HA | This paper | |
| NIX-RPE-AAA | This paper | |
| NIXD140-150 | This paper | |
| M71E_pLV_FBXL4_HA_WT_VB3 | This paper | |
| Arg544E_pLV_FBXL4_HA_WT_VB3 | This paper | |
| M71E and Arg544E_pLV_FBXL4_HA_WT_VB3 | This paper | |
| pLVX TetOne BNIP3 | This paper | |
| pLVX TetOne NIX | This paper | |
| Plasmid: pGEX-6P-2 | Cytiva | 28-9545-50 |
| Plasmid: pGEX-6P-2 GST-PPTC7(31-304) | This study | N/A |
| Plasmid: pGEX-6P-2 GST-PPTC7(31-304) (D290N) | This study | N/A |
| **Deposited Data** | | |
| SCF^βTRPC (crystal structure) | RSCB Protein Data Bank | PDB ID: 6TTU |
| Human PPTC7 (protein sequence) | Uniprot | Q8NI37 |
| Human FBXL4 (open reading frame) | Uniprot | Q9UKA2 |
| Human BNIP3 (protein sequence) | Uniprot | Q12983 |
| Human NIX (protein sequence | Uniprot | O60238 |
| **Software** | | |
| Pymol | Schrodinger, USA | https://pymol.org/2/ |
| Consurf | Simpson et al, 2021 | https://consurf.tau.ac.il/consurf_index.php |
| AlphaFold2 Multimer | Sowter et al, 2001; Stojanovski et al, 2007 | https://github.com/deepmind/alphafold |
| ColabFold and ColabFold batch | Sun et al, 2024 | https://github.com/sokrypton/ColabFold |
| **Other** | | |
| HiLoad™ Superdex75 PG | GE Healthcare | Catalogue: 28989333 |

## CRISPR-CAS9-mediated genome editing

To generate HeLa and U2OS PPTC7 KO cell lines, PPTC7-1_pLentiCRISPR V2 and PPTC7-2_pLentiCRISPR V2 plasmids were generated by Genscript® based on the the following gRNA sequences: TTCGTACCTAGTAATCCCAT and CGGCGACTACGGACTGGT GA, respectively. The pLenti-CRISPR V2 plasmids were used to generate lentiviruses in 293T cells. HeLa or U2OS cells were transduced with lentivirus, and ~24 h post-transduction, cells were selected with puromycin for 48 h. Successful knockout was confirmed using immunoblotting using antibodies to PPTC7. FBXL4-deficient U2OS cells (clone 2G10) have been described previously (Nguyen-Dien et al, 2023). PPTC7-deficient 293T cells have been described previously (Meyer et al, 2020).

## Cell culture, transfection and chemicals

Cell lines were incubated at 37 °C in a humidified incubator containing 5% $CO_2$. HeLa (ATCC CCL-2), U2OS (ATCC HTB-96) and HEK293T (ATCC CRL-3216) cells were propagated in Dulbecco's Modified Eagle's medium/Nutrient mixture F-12 GlutaMAX™ (DMEM/F-12; Thermo Fisher Scientific) supplemented with 10% foetal bovine serum (Gibco). All cell lines were regularly screened for mycoplasma contamination. Plasmid transfections were performed using Lipofectamine 2000 (Thermo Fisher Scientific) according to the manufacturer's recommendations. Cells were transfected with plasmid DNA using Lipofectamine 2000 (Thermo Scientific) for ~48 h. The following chemicals from Sigma were used: cycloheximide (CHX; 100 µg/ml; 66-81-9), deferiprone (DFP; 1 mM; 379409), DMOG (0.5 mM; D3695) and echinomycin (10 nM; SML0477). MLN4924 (0.5 µM; 85923S) was obtained from Cell Signaling Technology. MG132 (10 µM; 474787) was purchased from Merck.

HeLa mt-Keima cell lines expressing dox-inducible FLAG-S tag NIX-wild-type and NIX mutants were generated as described previously (Pagan et al, 2015). Briefly, pcDNA5/FRT/TO (Thermo Fisher) constructs expressing NIX or variants were co-transfected with pOG44 into HeLa-T-rex Flp-in cells mt-Keima cells to generate inducible cell lines using Flippase (Flp) recombination target (FRT)/Flp-mediated recombination technology. Twenty-four hours post-transfection, cells were selected with Hygromycin B (400 µg/ml) for approximately 10 days. To induce expression, cells were treated with 0.5 µg/ml doxycycline (Sigma; 10592-13-9).

## Virus production and transduction

Lentiviruses (pLV constructs) or retroviruses (pCHAC-mt-mKeima) were packaged in HEK293T cells. The media containing lentiviral or retroviral particles were harvested 48 h later. Cells were transduced with virus along with 10 µg/mL polybrene (Sigma). Following transduction, cell lines were either sorted using FACS based on fluorescence (for mt-Keima) or selected with puromycin (for cells with pLV constructs).

## Biochemical techniques, immunoblotting and co-immunoprecipitation

Immunoblotting was performed as previously described (Nguyen-Dien et al, 2023). Cultured cells were harvested and lysed in SDS lysis buffer (50 mM Tris, 2% SDS) followed by heating at 95 °C for 15 min. Protein extracts were diluted in Bolt™ LDS Sample Buffer (Invitrogen™; B0008). Equal amounts of protein samples were separated using BOLT pre-cast 4–12% gradient gels (Invitrogen™) and transferred onto methanol-pretreated Immobilon®-P PVDF Membrane (0.45 µm pore size) (Merck; IPVH00010) using BOLT gel transfer cassettes and BOLT transfer buffer (Invitrogen™; BT0006). Membranes were blocked in 5% skim milk for 1 h at room temperature followed by overnight incubation at 4 °C with the indicated primary antibodies. Chemiluminescent detection of HRP-conjugated secondary antibodies was performed using Pierce ECL Western blotting substrate (Thermo Fisher Scientific; 32106) or Pierce SuperSignal West Femto Substrate (Thermo Fisher Scientific; 34094) and ChemiDoc™ Imaging System (Bio-Rad). Phos-tag Precast gels were purchased from Fujifilm/WAKO and run according to the manufacturer's instructions. For immunoprecipitation, cellular lysis was performed using a Tris-Triton lysis buffer (50 mM Tris-Cl pH 7.5, 150 mM NaCl, 10% glycerol, 1 mM EDTA, 1 mM EGTA, 5 mM $MgCl_2$, 1 mM β-glycerophosphate, and 1% Triton), supplemented with protease inhibitor cocktail (Rowe Scientific; CP2778) and PhosSTOP EASYpack Phosphatase Inhibitor Cocktail (Roche; 4906837001), and kept on ice for 30 min. Subsequently, cell lysates were centrifuged at 21,130×g for 10 min at 4 °C. For immunoprecipitation of exogenously expressed FLAG-tagged or HA-tagged proteins, the cell lysates were incubated with bead-conjugated FLAG (Sigma; A2220) or bead-conjugated HA (Thermo Fisher Scientific; 88837), respectively, in a rotating incubator for 1–2 h at 4 °C. The immunoprecipitated complexes were then washed five times with Tris-Triton lysis buffer before elution with Bolt™ LDS Sample Buffer for subsequent immunoblotting.

## Protease import protection assay

Crude mitochondria were prepared by resuspending cell pellets in mitochondrial isolation buffer (250 mM mannitol, 0.5 mM EGTA and 5 mM HEPES–KOH pH 7.4) followed by homogenisation using a 26.5 G needle (303800, Becton Dickinson) for 10 strokes, as in (Adriaenssens et al, 2023). The homogenate was then centrifuged twice at 600×g for 10 min at 4 °C to remove cell debris and intact nuclei. The supernatant was then centrifuged twice at 7000×g for 10 min at 4 °C to acquire a mitochondrial pellet. The resuspended pellets were centrifuged twice at 10,000×g for 10 min at 4 °C. Isolated mitochondria were then divided into untreated or proteinase K-treated conditions (10 µg/ml Proteinase K, EO0492, Life Technologies). Samples were rotated for 20 min at 4 °C. Proteinase K digestion was then blocked with 1 mM phenylmethyl-sulfonyl fluoride (PMSF, sc-482875, Santa Cruz Biotechnology) on ice for 10 min. Samples were centrifuged at 10,000×g for 10 min at 4 °C. Pellets were resuspended in 1× NuPAGE LDS sample buffer (NP0007, Life Technologies) and analysed by immunoblotting.

## Protein expression and purification

GST-PPTC7 (31-304) in the pGEX-6P-2 vector was transformed into BL21(DE3) cells and plated on LB-Agar plates containing 0.1 mg/mL ampicillin. A single colony from this plate was used to inoculate 200 mL of LB medium (containing 0.1 mg/mL ampicillin) and grown overnight at 37 °C with shaking at 180 rpm. The following day, 10 mL of overnight culture was used to inoculate 1 L of LB medium, supplemented with 0.1 mg/mL ampicillin. Cultures were grown at 37 °C with shaking at 190 rpm until the $OD_{600}$ reached 0.9–1.0, at which point protein expression was induced by adding 0.5 mM IPTG and the temperature was lowered to 20 °C. Cultures continued to grow overnight. Cells were harvested the following day by centrifugation at 6200 rpm for 7 min using a Beckman JLA 8.1000 rotor. Cells were resuspended in lysis buffer (50 mM Tris, pH 8.0, 500 mM NaCl, 10% glycerol, benzamidine (0.1 mg/mL) and DNase (0.1 mg/mL) and lysed using a Cell Disruptor (Constant Systems) at 35 kPsi. The cell lysate was centrifuged at 14,000 rpm for 40 min at 4 °C using a Beckman JLA 16.250 rotor. The soluble fraction was then purified by affinity chromatography using glutathione-sepharose resin. The GST tag was removed from the protein by incubating the resin with PreScission protease overnight at 4 °C. Untagged PPTC7 was eluted the following day in lysis buffer. The elution was further purified by gel filtration chromatography using a Superdex 75 (16/600) column

(GE Healthcare) into buffer containing 50 mM HEPES (pH 7.5), 300 mM NaCl and 2 mM β-mercaptoethanol.

## Isothermal titration calorimetry (ITC)

All ITC experiments were performed using a Microcal PEAQ-ITC instrument (Malvern) at 25 °C. Before performing the metal-binding experiments, PPTC7 was incubated with 1 mM EDTA and then gel-filtered using a PD10 desalting column (Cytiva) to remove any bound metal ions. All metal-binding experiments were performed in buffer containing 50 mM HEPES (pH 7.5), 200 mM NaCl and 2 mM β-mercaptoethanol. In these experiments, 150 μM $MnCl_2$ or $MgCl_2$ was titrated into 15 μM PPTC7. Experiments to test the binding of PPTC7 to BNIP3 and NIX peptides were performed in buffer containing 50 mM HEPES (pH 7.5), 300 mM NaCl, 10 mM $MnCl_2$ and 2 mM β-mercaptoethanol. Peptide sequences were as follows: >BNIP3(N113-R139)NSDWIWDWS**SRPE**NIPPKEFLFKHPKR and >BNIP3L(A138-R163) ADWVSDWS**SRPE**NIPPKEFHFRHPKR. In all experiments, 500 μM of peptide was titrated into 25 μM PPTC7. The dissociation constant ($K_D$), enthalpy of binding (ΔH) and stoichiometry (N) were calculated using the MicroCal PEAQ-ITC software.

## pNPP phosphatase assay

The generic phosphatase substrate para-nitrophenyl phosphate (pNPP) was used to determine the phosphatase activity of recombinant PPTC7 wild-type and PPTC7-D290N mutant. In total, 10 mM pNPP (New England Biolabs) was added to 4 μM PPTC7 in the presence or absence of either 5 mM $MnCl_2$ or 5 mM $MgCl_2$. All reactions were carried out in buffer containing 50 mM HEPES (pH 7.5), 200 mM NaCl and 2 mM β-mercaptoethanol, in a total volume of 100 μL. The reaction was incubated for 15 min at room temperature before measuring the absorbance at 405 nm using an Infinite M1000 Pro plate reader (Tecan).

## Protein structural prediction, modelling and visualisation

All protein models were generated using AlphaFold2 Multimer (preprint: Evans et al, 2022; Jumper et al, 2021) implemented in the Colabfold interface available on the Google Colab platform (Mirdita et al, 2022). For each modelling experiment, ColabFold was executed using default settings where multiple sequence alignments were generated with MMseqs2 (Mirdita et al, 2019). For all final models displayed in this manuscript, structural relaxation of peptide geometry was performed with AMBER (Hornak et al, 2006). For all modelling experiments, we assessed (i) the prediction confidence measures (pLDDT and interfacial iPTM scores), (ii) the plots of the predicted alignment errors (PAE) and (iii) backbone alignments of the final structures. The model of PPTC7 and NIX bound to the SCF^FBXL4 complex was constructed in two steps. Firstly, the trimeric PPTC7–NIX–FBXL4 complex was predicted with AlphaFold2 as described above. This was then aligned to the structure of SCF^βTRCP consisting of the proteins βTRCP–Skp1–Cul1–Rbx1–Ube2d–Nedd8–Ub (PDB ID 6TTU) (Baek et al, 2020). All structural images were made with Pymol (Schrodinger, USA; https://pymol.org/2/).

## Indirect immunofluorescence staining and mt-Keima assay

Cells grown as monolayers on coverslips were fixed with ice-cold methanol. Cells were blocked with 2% BSA in PBS. Cells were then sequentially labelled with primary antibodies for 1 h, followed by the species-specific secondary antibodies for 1 h. Coverslips were mounted on glass microscope slides using Prolong Diamond Antifade Mountant (Thermo Fisher Scientific; P36965). Images were acquired using either a DeltaVision Elite inverted microscope system (GE Healthcare) or a using a ×60/1.4NA Oil PSF Objective from Olympus or Zeiss LSM900 Fast AiryScan2 Confocal microscope with a 63× C-Plan Apo NA 1.4 oil-immersion objective. DeltaVision images were processed using the Softworx deconvolution algorithm whereas Airyscan images were processed using ZEN Blue 3D software (version 3.4).

The mt-Keima assay was performed as previously described (Sun et al, 2017). A Leica DMi8 SP8 Inverted confocal microscope equipped with a ×63 Plan Apochromatic objective and environmental chamber (set to 5% $CO_2$ and 37 °C) was used to capture images. Quantitative analysis of mitophagy with mt-Keima was performed using Image J/Fiji software. Individual cells were isolated from the field of view by creating regions of interest (ROI). The chosen ROI were then cropped and separated into distinct channels before undergoing threshold processing. The fluorescence intensity of mt-Keima at 561 nm (indicating lysosomal signal) and 458 nm (indicating mitochondrial signal) at the single-cell level was measured, and the ratio of 561 nm to 458 nm was calculated. Three biological replicates were performed for each experiment, with >50 cells analysed per condition for each repeat.

## Statistical analysis

GraphPad Prism 9.0 software was used to perform statistical comparisons. The centre line and error bars on the graphs represent the mean and standard deviation of normalised biologically independent replications. Unless otherwise noted, three or more biologically independent replications for used for statistical comparisons. No blinding or randomisation was incorporated into the experimental design. *P* values greater than 0.05 were considered non-significant.

# Data availability

This study includes no data deposited in external repositories.

The source data of this paper are collected in the following database record: biostudies:S-SCDT-10_1038-S44319-024-00181-y.

# Peer review information

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

## Acknowledgements

The authors thank Natalie Niemi and David Pagliarini for helpful discussions, the FLAG-PPTC7 plasmid, and the PPTC7 KO 293T cells. Imaging was performed at the Microscopy and Image Analysis Facility in the School of Biomedical Sciences at the University of Queensland. Lentiviruses were produced by the University of Queensland (UQ)-Viral Vector Core. This work was supported by the Australian National Health and Medical Research Council (APP1183915 and APP1136021), a Brain Foundation Research grant (2020), a Mito Foundation Incubator Grant (2022), Mito Foundation Research Fellowship to PGK and a Mito Foundation Scholarship Top-up to KLK and an Australian Research Council Future Fellowship (FT180100172) to JKP, a Rebecca Cooper Foundation Fellowship (RC20241396) to ML, and an Australian Research Council Discovery Project (DP210102704) to MJKJ. This work was partially supported by NIH GM136250 to MP. MP is an investigator with the Howard Hughes Medical Institute. BMC is supported by an Australian NHMRC Investigator Grant (APP2016410). DK is supported by an Australian NHMRC Investigator Grant (GNT1178122).

## Author contributions

**Giang Thanh Nguyen-Dien**: Formal analysis; Validation; Investigation; Writing—review and editing. **Brendan Townsend**: Formal analysis; Validation; Investigation; Writing—review and editing. **Prajakta Gosavi Kulkarni**: Supervision; Validation; Investigation, Writing—review and editing. **Keri-Lyn Kozul**: Formal analysis; Validation; Investigation, Writing—review and editing. **Soo Siang Ooi**: Investigation. **Denaye N Eldershaw**: Investigation. **Saroja Weeratunga**: Investigation. **Meihan Liu**: Investigation. **Mathew JK Jones**: Conceptualisation; Supervision. **S Sean Millard**: Supervision; Writing—review and editing. **Dominic CH Ng**: Supervision; Writing—review and editing. **Michele Pagano**: Writing—review and editing. **Alexis Bonfim-Melo**: Methodology. **Tobias Schneider**: Visualisation. **David Komander**: Conceptualisation; Writing—review and editing. **Michael Lazarou**: Writing—review and editing. **Brett M Collins**: Conceptualisation; Supervision; Investigation; Visualisation; Writing-original draft; Writing—review and editing. **Julia K Pagan**: Conceptualisation; Supervision; Funding acquisition; Validation; Investigation; Writing—original draft; Writing—review and editing.

Source data underlying figure panels in this paper may have individual authorship assigned. Where available, figure panel/source data authorship is listed in the following database record: biostudies:S-SCDT-10_1038-S44319-024-00181-y.

## Disclosure and competing interests statement

DK is founder, shareholder and member of the SAB of Entact Bio and Proxima Bio. MP is a scientific cofounder of SEED Therapeutics; received research funding from and is a shareholder in Kymera Therapeutics; and is a consultant for, a member of the scientific advisory board of, and has financial interests in CullGen, SEED Therapeutics, Triana Biomedicines, and Umbra Therapeutics. However, no research funds were received from these entities, and the findings presented in this manuscript were not discussed with any person in these companies. ML is a member of the scientific advisory board and co-founder of Automera. The remaining authors declare no competing interests.

# Expanded View Figures

**Figure EV1.  PPTC7 is required for the FBXL4-mediated destabilization of NIX/BNIP3.**

(**A**) Distinct guide RNAs targeting PPTC7 result in BNIP3 and NIX upregulation. Cells were transfected PPTC7 guide 1 (G1) or guide 2 (G2). (**B**) BNIP3 and NIX are stabilised in PPTC7-deficient 293T cells. Cells were treated with cycloheximide for the indicated times before immunoblotting as indicated. (**C**) Inhibition of HIF1α with echinomycin does not prevent the accumulation of BNIP3 and NIX in PPTC7-deficient cells. U2OS cells or PPTC7 KO cells were treated with echinomycin for 24 h. Echinomycin completely prevented the DFP-induced upregulation of BNIP3 and NIX, but only partially prevented the accumulation of BNIP3 and NIX in PPTC7 KO cells. DFP induces the 32 kDa form of PPTC7. (**D**) FBXL4 requires PPTC7 for its ability to promote BNIP3 and NIX turnover. FBXL4-HA was expressed in parental, FBXL4 KO, PPTC7 KO, and FBXL4/PPTC7 dKO cells. BNIP3 and NIX protein levels were monitored by Western blotting in response to FBXL4 expression. (**E**) PPTC7-mediated downregulation of BNIP3 and NIX does not occur in FBXL4-deficient cells. PPTC7-FLAG was transfected into either PPTC7 KO or FBXL4 KO cells. Cells were fixed and stained for FLAG(PPTC7) (green) and either NIX or BNIP3 (magenta). The orange dotted line surrounds the cells that have been transfected with PPTC7. (**F**) PPTC7(HA) overexpression causes the downregulation of BNIP3 and NIX in U2OS and 293T cells in basal conditions and after DFP treatment. PPTC7(HA) was transduced into U2OS or 293T cells and the levels of BNIP3 and NIX were monitored by immunoblotting. (**G**) PPTC7 overexpression results in the downregulation of BNIP3 and NIX in basal conditions as well as after DFP treatment. U2OS cells or U2OS cells stably transfected with PPTC7(HA) were treated with DFP for 24 h. Cells were subjected to cycloheximide chase. s.e. = shorter exposure. (**H**) PPTC7 overexpression suppresses DFP-induced mitophagy. Mitophagy was assessed U2OS mt-Keima cells or U2OS mt-Keima cells overexpressing PPTC7(HA) in the presence or absence of DFP. Emission signals at neutral pH were obtained after excitation with the 458 nm laser (green), and emission signals at acidic pH were obtained after excitation with the 458 nm laser 561 nm laser (magenta). Mitophagy is represented as the ratio of mt-Keima 561 nm fluorescence intensity divided by mt-Keima 458 nm fluorescence intensity for individual cells normalised to the mean of the untreated U2OS cells. Translucent grey dots represent measurements from individual cells. Coloured circles represent the mean ratio from independent experiments. The centre lines and bars represent the mean of the independent replicates $+/-$ standard deviation. $P$ values were calculated based on the mean values using a one-way ANOVA. ****$P<0.0001$. $n = 3$ independent experiments. Data Information: (**E, H**) Scale bar = 20 microns. (**B, C**) The red asterisks indicate PPTC7-specific bands and 28 kDa, 32 kDa, and 40 kDa. Arrow and ns=non-specific band at ~36 kDa.

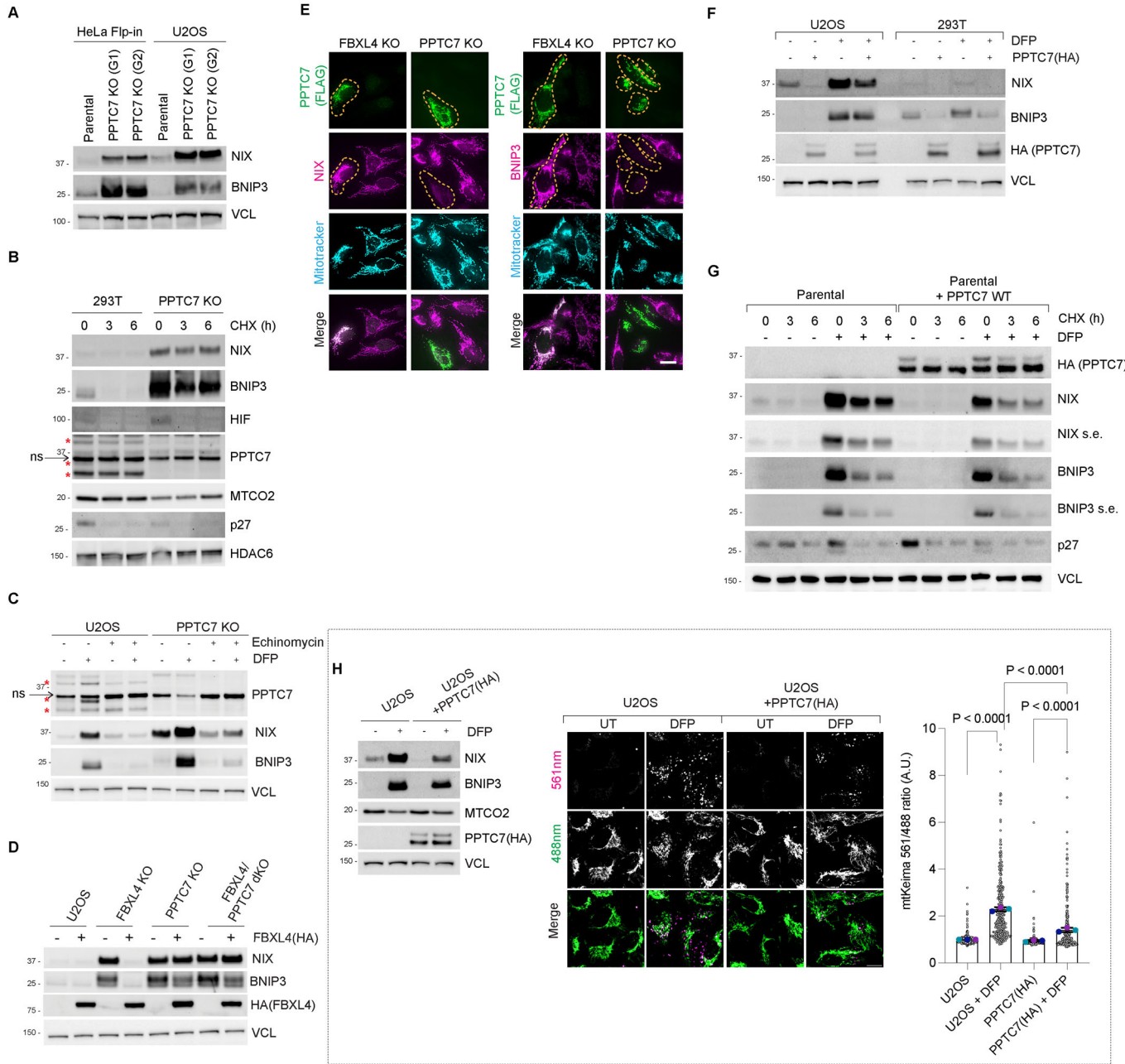

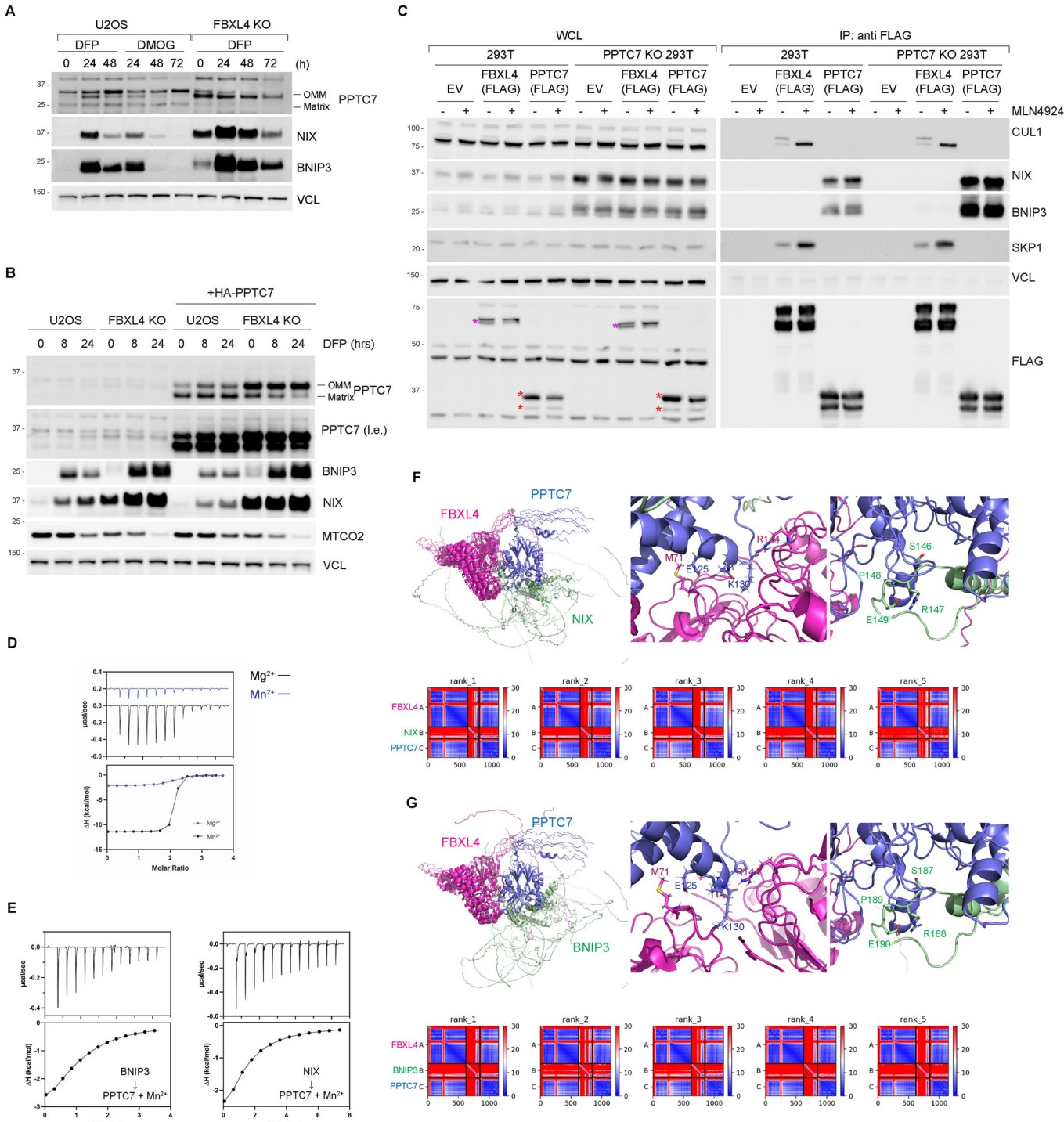

**Figure EV2. PPTC7 interacts with NIX/BNIP3 and FBXL4 and is not required for FBXL4 to interact with CUL1 and SKP1.**

(A) Analysis of the stability of the outer membrane form of PPTC7 in response to DFP treatment, DMOG treatment, and/or FBXL4 deficiency. (B) Analysis of the stability of the outer membrane form of endogenous PPTC7 and exogenous PPTC7(HA) in response to DFP treatment. (C) PPTC7 is not required for FBXL4 to interact with CUL1 or SKP1. 293T or 293T PPTC7 KO cells were transfected with either FLAG-FBXL4 or FLAG-PPTC7. Cells were treated with MLN4924 for 24 h where indicated. Cell lysates were immunoprecipitated with anti-FLAG beads, and the immuno-precipitates were analysed by immunoblotting as indicated. FBXL4 binds to CUL1 and SKP1, whereas PPTC7 binds to BNIP3 and NIX. WCL = whole-cell lysates. Red asterisks mark the PPTC7 transfected product and magenta asterisks mark the FBXL4 transfected product. (D) ITC comparison of PPTC7 wild-type binding to $Mg^{2+}$ and $Mn^{2+}$. The binding affinities were 6.21 nM for $Mn^{2+}$ and 237 nM for $Mg^{2+}$. (E) ITC comparison of PPTC7 wild-type binding to BNIP3 and NIX in the presence of $Mn^{2+}$. The binding affinities were 20.1 µM for BNIP3 and 37.5 µM for NIX. (F) Overlay of the top 5 AlphaFold2 models of FBXL4, PPTC7, NIX with predicted aligned error (PAE) plots. (G) Overlay of the top 5 AlphaFold2 models of FBXL4, PPTC7, BNIP3 with PAE plots.

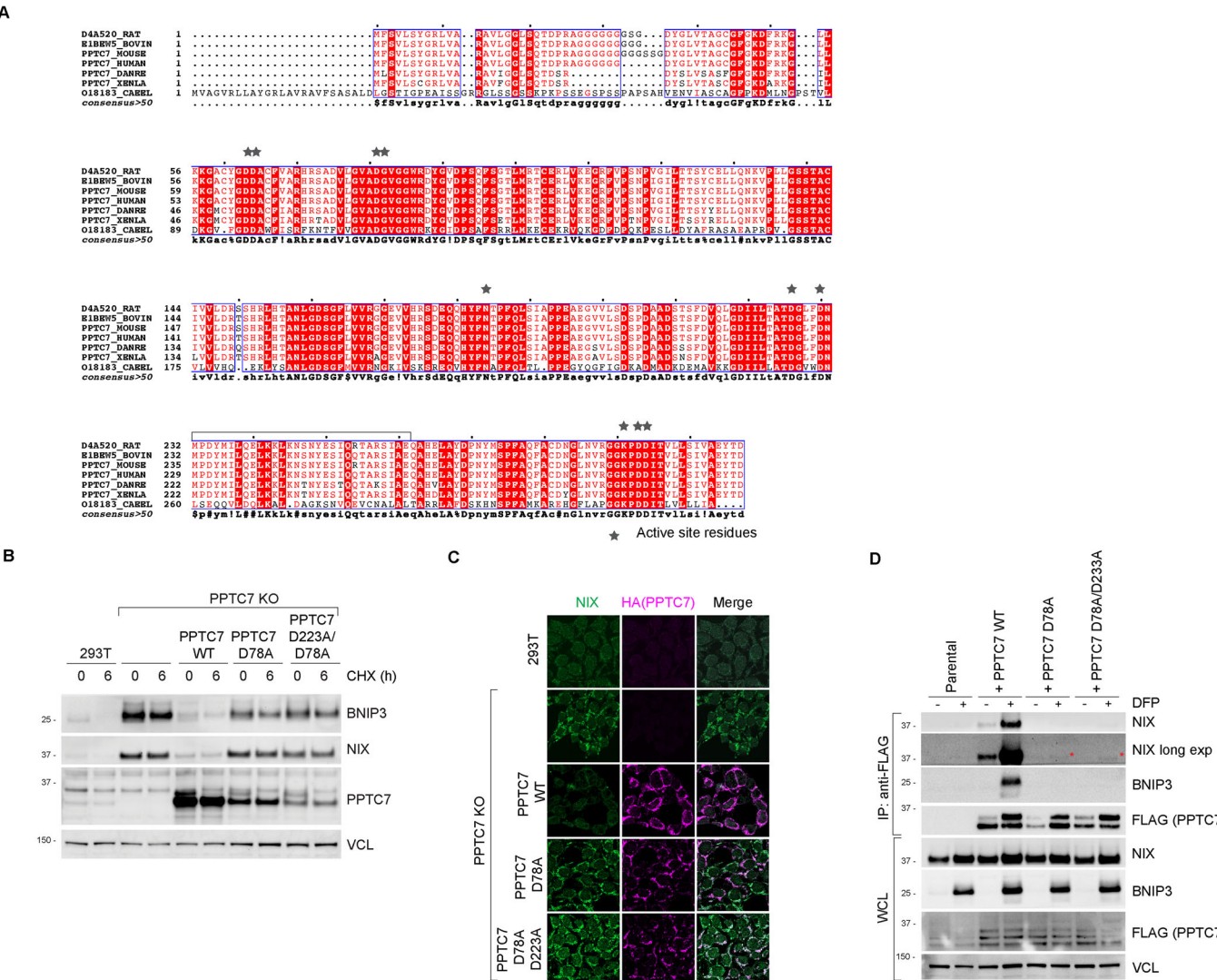

**Figure EV3.  Radical disruption of PPTC7's catalytic site interferes with its binding to BNIP3 and NIX.**

(A) Sequence alignment of PPTC7 orthologues with active site residues from Fig. 3A indicated. (B) Disruption of the PPTC7's active site residues from aspartate to alanine compromises its ability to downregulate BNIP3 and NIX. PPTC7 KO cells were transduced with PPTC7 wild-type, PPTC7-D78A, or PPTC7-D223A/D78A. Wild-type PPTC7 rescued the turnover of BNIP3 and NIX, however, the D78A and D223A/D78A variants did not. (C) Aspartate to alanine mutations in PPTC7's active cannot rescue the downregulation of NIX by PPTC7. PPTC7 KO cells were complemented with PPTC7 or active site mutants and NIX levels were analysed by immunofluorescence microscopy. Scale bar = 20 microns. (D) Disruption of the PPTC7's active site residues from aspartate to alanine interferes with its ability to bind to BNIP3 and NIX. Cell lysates expressing PPTC7(FLAG) and mutants were immunoprecipitated with anti-FLAG beads, and the immuno-precipitates were analysed by immunoblotting as shown. Unlike the PPTC7-D78N mutant in Fig. 3B, the D78A mutant is unable to bind to BNIP3 or NIX.

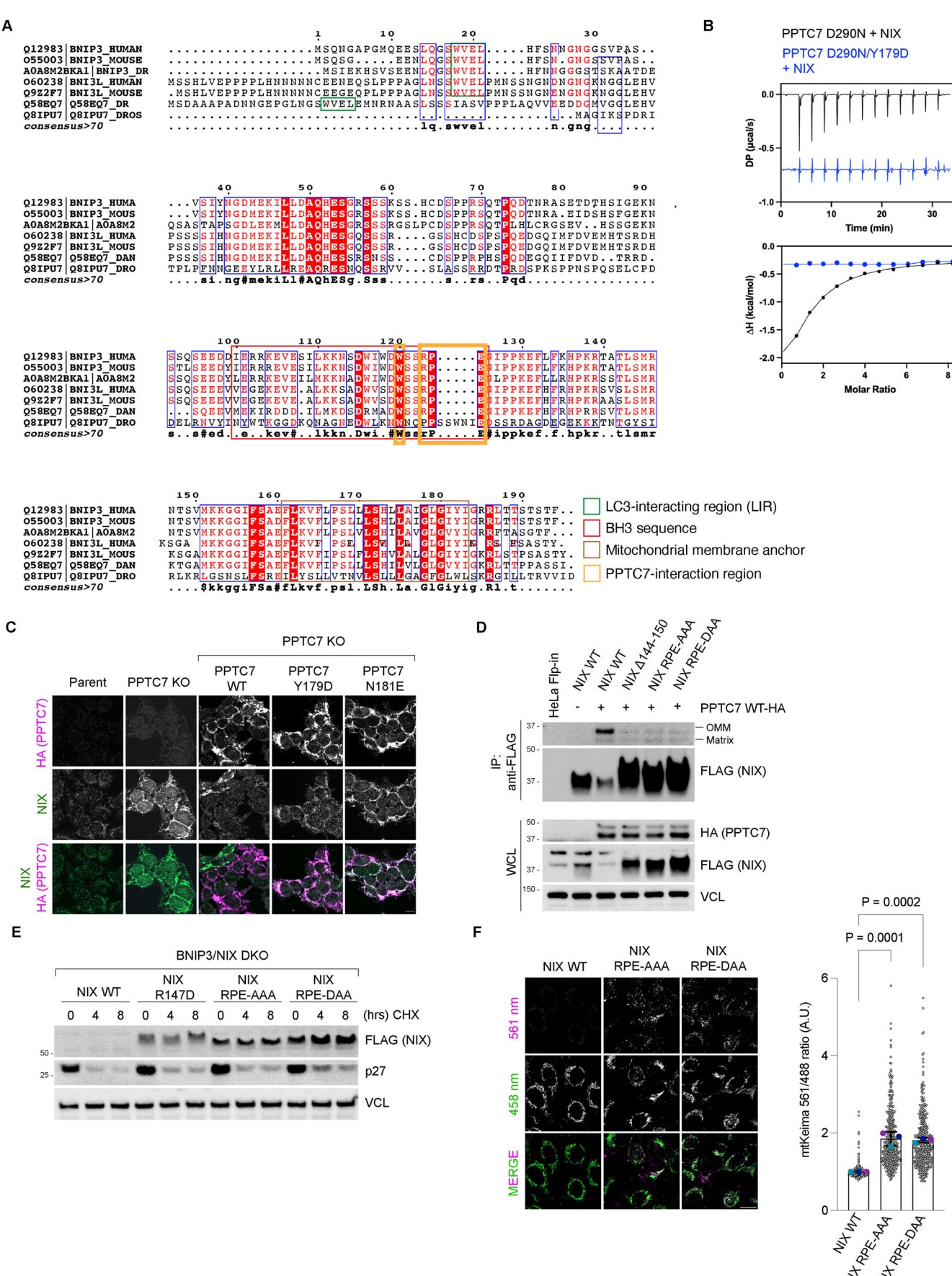

**Figure EV4.   The NIX-PPTC7 interaction is critical for NIX turnover and mitophagy suppression.**

(A) Sequence alignment of BNIP3 and NIX orthologues. Functionally relevant motifs or domains are indicated. BNIP3 accession Q12983 (194 aa) replaces the previous BNIP3 accession EAW49143.1 (259 aa) characterised previously (Nguyen-Dien et al, 2023). (B) ITC comparison of PPTC7-D290N and PPTC7-D290N/Y179D binding to NIX peptide. The binding affinities were 35.9 ± 1.09 for PPTC7-D290N and non-binding for PPTC7-D290N/Y179D. (C) PPTC7-Y179D and PPTC7-N181E variants localise to mitochondria. Wild-type PPTC7 can reduce NIX levels when expressed in PPTC7 KO cells, but PPTC7-Y179D and PPTC7-N181E cannot. (D) Arg147 in NIX is critical for binding to PPTC7. PPTC7(HA) was transduced into cell lines expressing inducible NIX mutants. NIX expression was induced with doxycycline for 24 h. Cell lysates were immunoprecipitated with anti-FLAG beads, and the immuno-precipitates were analysed by immunoblotting. (E) Arg147 in NIX is critical for its turnover. HeLa Flp-in BNIP3/NIX double KO cells expressing FLAG-tagged NIX-WT or NIX binding mutants (FLAG-tagged NIX Δ144-150, NIX-RPE-AAA and NIX-RPE-DAA) were subjected to a cycloheximide chase. (F) Expression of NIX-RPE/AAA and NIX-RPE/DAA in NIX leads to an increase in basal levels of mitophagy compared with NIX-wildtype. Hela Flp-In NIX knockout/ Hela Flp-In BNIP3/NIX double knockout Keima cells stably expressing NIX mutants were treated with doxycycline for 48 h and mitophagy was evaluated using live-cell confocal fluorescence microscopy. Translucent grey dots represent measurements from individual cells. Coloured circles represent the mean ratio from independent experiments. The centre lines and bars represent the mean of the independent replicates +/− standard deviation. *P* values were calculated based on the mean values using a one-way ANOVA. Data Information: (C, F) Scale bars = 20 microns.

