## [Peer Review File · EMBO Reports]

PPTC7 antagonizes mitophagy by promoting BNIP3 and NIX degradation via SCFFBXL4

Giang Thanh Nguyen-Dien, Brendan Townsend, Prajakta Kulkarni, Keri-Lyn Kozul, Soo Siang Ooi, Denaye Eldershaw, Saroja Weeratunga, Meihan Liu, Mathew JK Jones, S Sean Millard, Dominic CH Ng, Michael Lazarou, Michele Pagano, David Komander, Brett M Collins, Julia Pagan, Tobias Schneider, and Alexis Bonfim-Melo

Corresponding author(s): Julia Pagan (j.pagan@uq.edu.au) , Brett Collins (b.collins@imb.uq.edu.au)

Review Timeline:

Transfer Date:	21st Apr 24
Editorial Decision:	23rd Apr 24
Revision Received:	1st May 24
Editorial Decision:	14th May 24
Revision Received:	30th May 24
Accepted:	4th Jun 24

Transaction Report: A revised version of this manuscript was transferred to EMBO reports following peer review at the EMBO Journal.

Referee #1:

The authors previously showed a role for FBXL4 in suppressing mitophagy via turnover of BNIP3 and NIX (Nguyen-Dien et al, 2023). Here, the authors explore the interaction of FBXL4 with PPTC7 phosphatase in regulating BNIP3 and NIX since like FBXL4 inactivation, loss of PPTC7 also leads to decreased mitochondria, increased mitophagy and perinatal lethality in mice. Here they show that PPTC7 is rate-limiting for the ability of FBXL4 to promote turnover of BNIP3 and NIX. They identify two forms of PPTC7, a 32 kD OMM form and a 28 kD matrix form and show that OMM PPTC7 interacts with BNIP3 and NIX, an interaction that is independent of FBXL4. Conversely, the interaction of FBXL4 with SKP1 and CUL1 is PPTC7-independent. Importantly, the authors map the sequences in BNIP3 and NIX required for PPTC7 interaction to the conserved domains carboxy terminal to the non-canonical "BH3" domain in BNIP3/NIX where a SRPE motif is particularly critical for binding to PPTC7. Also critically here the authors mutate PPTC7 in a way that inhibited its phosphatase activity without preventing interaction with BNIP3 and NIX to show that the phosphatase activity of PPTC7 is not required for the ability of FBXL4 to promote degradation of BNIP3 and NIX. This is consistent with similar work performed by Sun et al, 2024. Finally, they map the residues required for the FBXL4 interaction with PPTC7 and show that this interaction is required for the ability of FBXL4 to promote proteasomal turnover BNIP3 and NIX.

Overall, the work is convincing and well presented with the one concern regarding novelty given the recent work from Sun et al, 2024 in Molecular Cell. The authors could have investigated further what causes the PPTC7 32 kD OMM form to accumulate which would have added a novel component here.

Major Points.

1. The authors claim that their work differs from Sun et al who propose a role for PPTC7 in recruiting CUL1 to the FBXL4 complex but they really did not test this here in their work.
2. The authors do not examine what causes OMM-PPTC7 to accumulate at the OMM which remains unanswered here or in Sun et al, 2024.

Minor Points.

1. In this paper and in the previous paper, the amino acid numbering for BNIP3 is incorrect. The SRPE motif referred to as amino acids 187 - 190 in BNIP3 should be amino acids 122 - 125. Numbering in NIX is correct. They have the numbering correct in Supplemental figure 4A, so it is unclear why they cite the wrong amino acids in the text. Sun et al, 2024 have the numbering correct for reference.

Referee #2:

In this study Nguyen-Dien and colleagues investigate regulation of BNIP3 and NIX mediated mitophagy, building from recent work by themselves that the Ub ligase complex SCF FBXL4 targets both proteins for degradation. They describe a role for the phosphatase PPTC7 in regulating this process, in essence acting as a bridging molecule between NIX, BNIP3 and FBXL4, facilitating the degradation of the NIX and BNIP3, thus counteracting mitophagy. Overall, the study is timely the data is robust and supports the authors' conclusions. I have some points for consideration during revision:

- Figure 1A - it is stated that loss of PPTC7 increases the half-life of BNIP3 and NIX however this is never quantified/demonstrated relative to WT cells (realise its challenging in the WT setting since especially BNIP3 levels are low. Would suggest the wording of this statement is changed somewhat.

- Figure 1 - Is there any effect of PPTC7 loss on BNIP3 and NIK transcript levels ? i.e. could this also contribute to high levels of NIK and/or BNIP3

- points for discussion, while the mitophagy probe does indicate increased mitophagy in the absence of PPTC7 and FBXL4, the effects are quite subtle, least throughout the level of mitochondrial protein content don't appear significantly less, some possibilities indicate additional levels of control of NIK, BNIP3 activity and/or compensatory increases in mitochondrial biogenesis.

Dear Julia,

Thank you for the transfer of your manuscript to EMBO Reports. As my colleague at The EMBO Journal, Dr. Hartmut Vodermaier, already told you, we would like to offer rapid publication of a revised version, responding to both referees' minor points but without extension towards referee 1's major points, unless you have data at hand that might address these concerns and that you want to include. In either case, please discuss the major concerns from referee 1 in the manuscript and in a point-by-point response.

I am happy to discuss the revision further by e-mail or video call, if you wish.

Please find the general formatting guidelines below my signature. In addition, I will list here some specific points that I kindly ask you to address, as it will speed up the quality control of your revised paper and ultimately its publication.

- Please provide the Reagents and Resource table as a separate Word file (file type "Reagents and Resource table" in the manuscript tracking system). It will be typeset into the article by the production team.
- The column headers are "Reagent/Resource", "Reference or Source", "Identifier or Catalog Number".
- You need a Data availability paragraph at the end of Materials and Methods.
- You need a 'Disclosure and competing interests statement'. For more information see <https://www.embopress.org/page/journal/14693178/authorguide#conflictsofinterest>
- Please provide up to 5 keywords.
- Please remove the DOIs from the reference list.
- The manuscript word file may not contain figures.
- In the figure legends, 'n' must be specified and also whether this refers to biological/independent or technical replicates. Fig 1H seems to base statistics on n = 1 with several cells quantified for each condition? If this is the case, please remove the statistical analysis.
- Please only define the p-values that are actually shown in the figure panel(s).
- If information refers to several panels in the figure (e.g., mean +/- SD, or 'n' or p-values or scale bars) then please summarize them in a "Data Information: [...]" section at the end of the legends incl. a reference to the panels this information applies to.
- Fig S1E, H, S3C, S4B, E, Fig 5D, E either lack scale bars or they are too small/thin to be seen.
- Fig S4E lacks information on 'n'
- Supplementary figures: You could upload them as EV figures (expandable in the html version). In that case the nomenclature is Figure EV# and their legend is part of the main manuscript with the header "Expanded View Figure Legends".
- We need an Author Checklist.
- You will be contacted by our Source Data coordinator, Hannah Sonntag, listing all figure panels for which we need the minimally processed source data for. This will include all panels showing Western blots, quantification, and imaging data. You might already want to prepare these data. The structure is: One folder per figure containing subfolders for each panel. The 'figure folder' is zipped and uploaded.
An alternative is the deposition at Biostudies, in which case you insert a link that resolves to the dataset in the Data availability section.

- We also need a synopsis image (550 pixels width, 400 - 600 pixels wide, jpg or png) and a summary text (1-2 sentences), plus 3-4 bullet points.

That should be the most important items. General formatting guidelines are below my signature.

I look forward to receiving the revised manuscript.

Kind regards,

Martina

Martina Rembold, PhD
Senior Editor
EMBO Reports

GENERAL FORMATTING GUIDELINES:

2) individual production quality figure files as .eps, .tif, .jpg (one file per figure).

Please download our Figure Preparation Guidelines (figure preparation pdf) from our Author Guidelines pages <https://www.embopress.org/page/journal/14693178/authorguide> for more info on how to prepare your figures.

4) a complete author checklist, which you can download from our author guidelines

(<<https://www.embopress.org/page/journal/14693178/authorguide>>). Please insert information in the checklist that is also reflected in the manuscript. The completed author checklist will also be part of the RPF.

5) Please note that all corresponding authors are required to supply an ORCID ID for their name upon submission of a revised manuscript (<<https://orcid.org/>>). Please find instructions on how to link your ORCID ID to your account in our manuscript tracking system in our Author guidelines

(<<https://www.embopress.org/page/journal/14693178/authorguide#authorshipguidelines>>)

6) We replaced Supplementary Information with Expanded View (EV) Figures and Tables that are collapsible/expandable online. A maximum of 5 EV Figures can be typeset. EV Figures should be cited as 'Figure EV1, Figure EV2' etc... in the text and their respective legends should be included in the main text after the legends of regular figures.

<<https://www.embopress.org/page/journal/14693178/authorguide#expandedview>>

7) Please note that a Data Availability section at the end of Materials and Methods is now mandatory. In case you have no data that requires deposition in a public database, please state so instead of referring to the database.

See also < <https://www.embopress.org/page/journal/14693178/authorguide#dataavailability>>. Please note that the Data Availability Section is restricted to new primary data that are part of this study.

Additional information on source data and instruction on how to label the files are available
<<https://www.embopress.org/page/journal/14693178/authorguide#sourcedata>>.

10) Figure legends and data quantification:

- the name of the statistical test used to generate error bars and P values,
 - the number (n) of independent experiments (please specify technical or biological replicates) underlying each data point,
 - the nature of the bars and error bars (s.d., s.e.m.)
- If the data are obtained from n {less than or equal to} 5, show the individual data points in addition to the SD or SEM.
- If the data are obtained from n {less than or equal to} 2, use scatter blots showing the individual data points.

11) Our journal encourages inclusion of *data citations in the reference list* to directly cite datasets that were re-used and obtained from public databases. Data citations in the article text are distinct from normal bibliographical citations and should directly link to the database records from which the data can be accessed. In the main text, data citations are formatted as follows: "Data ref: Smith et al, 2001" or "Data ref: NCBI Sequence Read Archive PRJNA342805, 2017". In the Reference list, data citations must be labeled with "[DATASET]". A data reference must provide the database name, accession number/identifiers and a resolvable link to the landing page from which the data can be accessed at the end of the reference. Further instructions are available at <<https://www.embopress.org/page/journal/14693178/authorguide#referencesformat>>.

12) All Materials and Methods need to be described in the main text. We would encourage you to use 'Structured Methods', our new Methods format. According to this format, the Methods section should include a Reagents and Tools Table (listing key reagents, experimental models, software and relevant equipment and including their sources and relevant identifiers) followed by a Methods and Protocols section in which we encourage the authors to describe their methods using a step-by-step protocol format with bullet points, to facilitate the adoption of the methodologies across labs. More information on how to adhere to this format as well as downloadable templates (.doc or .xls) for the Reagents and Tools Table can be found in our author guidelines: < <https://www.embopress.org/page/journal/14693178/authorguide#manuscriptpreparation>>.

<<https://www.embopress.org/doi/10.15252/msb.20178071>>.

13) As part of the EMBO publication's Transparent Editorial Process, EMBO Reports publishes online a Review Process File to accompany accepted manuscripts. This File will be published in conjunction with your paper and will include the referee reports, your point-by-point response and all pertinent correspondence relating to the manuscript.

Dear Dr Rembold,

Thank you for your positive feedback on the rapid publication following the transfer of our manuscript from the EMBO Journal.

Please find below our responses to the reviewer's comments. I have attached a word document with track changes so that you can see our edits to the original manuscript.

Kind regards,

Julia

Referee #1:

The authors previously showed a role for FBXL4 in suppressing mitophagy via turnover of BNIP3 and NIX (Nguyen-Dien et al, 2023). Here, the authors explore the interaction of FBXL4 with PPTC7 phosphatase in regulating BNIP3 and NIX since like FBXL4 inactivation, loss of PPTC7 also leads to decreased mitochondria, increased mitophagy and perinatal lethality in mice. Here they show that PPTC7 is rate-limiting for the ability of FBXL4 to promote turnover of BNIP3 and NIX. They identify two forms of PPTC7, a 32 kD OMM form and a 28 kD matrix form and show that OMM PPTC7 interacts with BNIP3 and NIX, an interaction that is independent of FBXL4. Conversely, the interaction of FBXL4 with SKP1 and CUL1 is PPTC7-independent. Importantly, the authors map the sequences in BNIP3 and NIX required for PPTC7 interaction to the conserved domains carboxy terminal to the non-canonical "BH3" domain in BNIP3/NIX where a SRPE motif is particularly critical for binding to PPTC7. Also critically here the authors mutate PPTC7 in a way that inhibited its phosphatase activity without preventing interaction with BNIP3 and NIX to show that the phosphatase activity of PPTC7 is not required for the ability of FBXL4 to promote degradation of BNIP3 and NIX. This is consistent with similar work performed by Sun et al, 2024. Finally, they map the residues required for the FBXL4 interaction with PPTC7 and show that this interaction is required for the ability of FBXL4 to promote proteasomal turnover BNIP3 and NIX.

Overall, the work is convincing and well presented with the one concern regarding novelty given the recent work from Sun et al, 2024 in Molecular Cell. The authors could have investigated further what causes the PPTC7 32 kD OMM form to accumulate which would have added a novel component here.

Major Points.

1. The authors claim that their work differs from Sun et al who propose a role for PPTC7 in recruiting CUL1 to the FBXL4 complex but they really did not test this here in their work.
2. The authors do not examine what causes OMM-PPTC7 to accumulate at the OMM which remains unanswered here or in Sun et al, 2024.

We appreciate the reviewer's feedback on our manuscript. We agree with the two main points raised and believe that addressing them in future work will provide valuable insights into the function and localization of PPTC7.

Minor Points.

1. In this paper and in the previous paper, the amino acid numbering for BNIP3 is incorrect. The SRPE motif referred to as amino acids 187 - 190 in BNIP3 should be amino acids 122 - 125. Numbering in NIX is correct. They have the numbering correct in Supplemental figure 4A, so it is unclear why they cite the wrong amino acids in the text. Sun et al, 2024 have the numbering correct for reference.

Thank you for highlighting this issue, ensuring we clarify it in the current manuscript. When we initially began our research on BNIP3 several years ago, we used a DNA sequence encoding a 259 amino acid (aa) protein, as described in our previous manuscript (Nguyen-Dien). We specified this in the methods section of that manuscript (EAW49143.1 (259 aa)). We have since switched to using the 194 aa version of BNIP3 since it has replaced the 259 aa version in all databases, and our recent experiments utilize this shorter construct (note that this paper only includes experiments involving transfected NIX).

To clarify this point, we have amended the figure legend for Figure S3a to:

“Sequence alignment of BNIP3 and NIX orthologues. Functionally relevant motifs or domains are indicated. BNIP3 accession Q12983 (194 aa) replaces the previous BNIP3 accession EAW49143.1 (259 aa) characterised previously¹⁸.”

Referee #2:

In this study Nguyen-Dien and colleagues investigate regulation of BNIP3 and NIX mediated mitophagy, building from recent work by themselves that the Ub ligase complex SCF FBXL4 targets both proteins for degradation. They describe a role for the phosphatase PPTC7 in regulating this process, in essence acting as a bridging molecule between NIX, BNIP3 and FBXL4, facilitating the degradation of the NIX and BNIP3, thus counteracting mitophagy. Overall, the study is timely the data is robust and supports the authors' conclusions. I have some points for consideration during revision:

Thank you to the reviewer for their comments.

- Figure 1A - it is stated that loss of PPTC7 increases the half-life of BNIP3 and NIX however this is never quantified/demonstrated relative to WT cells (realise its challenging in the WT setting since especially BNIP3 levels are low. Would suggest the wording of this statement is changed somewhat.

We agree that without quantification and because BNIP3 levels are so low in steady-state conditions, it is inaccurate to say that the half-life increases, therefore as suggested we have amended the text in line 97 to “levels” instead of “half-life”.

- Figure 1 - Is there any effect of PPTC7 loss on BNIP3 and NIX transcript levels ? i.e. could this also contribute to high levels of NIK and/or BNIP3

We did not directly test the transcript levels of BNIP3 and NIX, however both Sun et al (Mol Cell, 84, 327, Figure 1E) and Wei et al (10.1101/2024.01.24.576953, Figure 1D) demonstrate that PPTC7 knockout does not affect BNIP3 and NIX transcript levels. We have now cited this result and referred to post-transcriptional regulation of BNIP3 and NIX in the text in line 367.

- points for discussion, while the mitophagy probe does indicate increased mitophagy in the absence of PPTC7 and FKBL4, the effects are quite subtle, least throughout the level of mitochondrial protein content don't appear significantly less, some possibilities indicate additional levels of control of NIK, BNIP3 activity and/or compensatory increases in mitochondrial biogenesis.

We have included the following text in the discussion:

Our findings show that the up-regulation of BNIP3 and NIX due to PPTC7 or FBXL4 disruption leads to mitophagy in only a subset of mitochondria. This suggests that while the increased expression of BNIP3 and NIX is necessary to trigger mitophagy, additional factors are required for its full induction, or that mitochondrial biogenesis compensates for the increased mitophagy.

Dear Julia,

Thank you for the submission of your revised manuscript to EMBO Reports.

We have completed all checks from the editorial side and I would kindly ask you to address the points below before we can proceed with the official acceptance.

Thank you very much and please do not hesitate to contact me in case you have any questions.

Kind regards,

Martina

Points to address:

- 1) Please make sure to cite the related manuscript from Lianjie Wei, Natalie Niemi and colleagues.
- 2) We noticed the following author name discrepancies:
 - Giang Thanh Nguyen-Dien in the manuscript vs. Giang Nguyen-Dien in the online manuscript tracking system;
 - Mathew JK Jones in the manuscript vs. Mathew Jones in the system;
 - S Sean Millard in the manuscript vs. Sean Millard in the system;
 - Dominic CH Ng in the manuscript vs. Dominic Ng in the system;
 - Julia K Pagan in the manuscript vs. Julia Pagan in the system.Please ensure that the information is correct and matches.
- 3) You state in the Author Checklist that the cell lines were tested for mycoplasma contamination. Please include this information also in the Methods section.
- 4) The Author Checklist will be published together with the Review Process File. Please complete the information on Corresponding Author Name, Journal Submitted to, and Manuscript Number in the file.
- 5) Please ensure that the funding information in the manuscript tracking system is complete, as this is the information that will be transferred to our publisher and to PubMed. In this respect we note that the Brain Foundation Research grant (2020) is missing in the online system.
- 6) The following figure panels are never called out in the manuscript text: Figure 2BCD and Figure 6AB. Please add callouts where appropriate.
- 7) There are callouts to Figure 4H and EV Tables 1-3 but these are either not present in the figure (Fig 1) or missing (EV tables). This needs to be rectified.
- 8) You mention figures in Appendix 1A and Appendix 1B in the legend of Figure 1 but the corresponding files are missing. Please either provide these figures or correct the callouts in case you refer to an EV figure.
- 9) The manuscript sections should be in the following order: Title page - Abstract & Keywords - Introduction - Results - Discussion - Methods - Data Availability - Acknowledgments - Disclosure Statement & Competing Interests - References - Figure Legends - (Tables with legends) - Expanded View Figure Legends.
- 10) Preprint citations: Please add the prefix preprint: to the in-text citation and [PREPRINT] in the reference list. I.e., in-text citation (preprint: NAME1 et al, YEAR) and in reference list Author NAME1, Author NAME2, (YEAR) article title. bioRxiv doi: nnn [PREPRINT]
See also <https://www.embopress.org/page/journal/14693178/authorguide#referencesformat>
- 11) Our production/data editors have asked you to clarify several points in the figure legends (see below). Please incorporate these changes in the manuscript and return the revised file with tracked changes with your final manuscript submission.
 - Please note that the legend for figure 5e is incorrectly labelled as 5i in the manuscript. This needs to be rectified.
 - Please note that the legend for figure EV 4f is incorrectly labelled as EV 4h in the manuscript. This needs to be rectified.
 - Please note that the exact p value is not provided in the legends of figures 4g; 5e; EV 1h; EV 4h.
 - Please note that in figure 4g; there is a mismatch between the annotated p values in the figure legend and the annotated p values in the figure file that should be corrected.
 - Please note that information related to n is missing in the legend of figure 3f.

- Although 'n' is provided, please describe the nature of entity for 'n' in the legend of figure 5e.
- Please note that the error bars are not defined in the legend of figure 3f.
- Please note that the scale bar needs to be defined for figures 3d; 5d; EV 1e, h; EV 3c; EV 4c, f.
- Please note that the red asterisk is not defined in the legend of figure 1c-d, g; 2b-e; 5c; EV 1b-c; EV 2c; EV 3d. This needs to be rectified.
- Figure 3F: please show the individual datapoints in addition to the mean

12) It seems that the e-mail address from co-author Soo Siang Ooi (s.ooi@uq.edu.au) is not active anymore, since our e-mail bounced. Please ensure that the e-mail address is up-to-date.

13) Source data: you should be contacted by our source data coordinator Hannah Sonntag. But please have the source data ready for upload, or alternatively, upload it to BioStudies and include a link in the Data Availability section. We will need the raw data (minimally processed) for all panels showing Western blots, immunofluorescence and quantifications (.xls files). The order is: One folder per figure with subfolders for each subpanel. This applies to all main figures, source data for EV figures is optional.

14) Finally, EMBO Reports papers are accompanied online by

A) a short (1-2 sentences) summary of the findings and their significance,

B) 2-3 bullet points highlighting key results and

C) a schematic summary figure that provides a sketch of the major findings (not a data image).

Please provide the summary figure as a separate file in PNG or JPG format at a size of 550x300-600 pixels (width x height).

Please note that the size is rather small and that text needs to be readable at the final size. Please send us this information along with the revised manuscript.

15) On a different note, I would like to alert you that EMBO Press offers a new format for a video-synopsis of work published with us, which essentially is a short, author-generated film explaining the core findings in hand drawings, and, as we believe, can be very useful to increase visibility of the work. This has proven to offer a nice opportunity for exposure i.p. for the first author(s) of the study. Please see the following link for representative examples and their integration into the article web page:

<https://www.embopress.org/doi/full/10.15252/embj.2019103932>

With kind regards,

All editorial and formatting issues were resolved by the authors.

Dr. Julia Pagan
University of Queensland
Otto Building
QLD 4067
Australia

Dear Julia,

I am very pleased to accept your manuscript for publication in the next available issue of EMBO reports. Thank you for your contribution to our journal.

Kind regards,

Martina
